# Signaling by a bacterial phytochrome histidine kinase involves a conformational cascade reorganizing the dimeric photoreceptor

E. Sethe Burgie[1,3], Katherine Basore [2], Michael J. Rau[2,3], Brock Summers [2], Alayna J. Mickles [1], Vadim Grigura[1], James A. J. Fitzpatrick[2,4] & Richard D. Vierstra[1] ✉

Phytochromes (Phys) are a divergent cohort of bili-proteins that detect light through reversible interconversion between dark-adapted Pr and photo-activated Pfr states. While our understandings of downstream events are emerging, it remains unclear how Phys translate light into an interpretable conformational signal. Here, we present models of both states for a dimeric Phy with histidine kinase (HK) activity from the proteobacterium *Pseudomonas syringae*, which were built from high-resolution cryo-EM maps (2.8–3.4-Å) of the photosensory module (PSM) and its following signaling (S) helix together with lower resolution maps for the downstream output region augmented by RoseTTAFold and AlphaFold structural predictions. The head-to-head models reveal the PSM and its photointerconversion mechanism with strong clarity, while the HK region is interpretable but relatively mobile. Pr/Pfr comparisons show that bilin phototransformation alters PSM architecture culminating in a scissoring motion of the paired S-helices linking the PSMs to the HK bidomains that ends in reorientation of the paired catalytic ATPase modules relative to the phosphoacceptor histidines. This action apparently primes autophosphorylation enroute to phosphotransfer to the cognate DNA-binding response regulator AlgB which drives quorum-sensing behavior through transient association with the photoreceptor. Collectively, these models illustrate how light absorption conformationally translates into accelerated signaling by Phy-type kinases.

All kingdoms of life employ a collection of photoreceptors to entrain their movement/growth, development, and reproduction to the ambient light environment. One prominent class are the phytochromes (Phy), a photoreceptor superfamily widely dispersed among bacteria, fungi, and plants that exploits a covalently-bound bilin (or open-chain tetrapyrrole) prosthetic group for light capture[1–4]. Through reversible photointerconversion between red-light-absorbing Pr and far-red light-absorbing Pfr conformers, they uniquely enable detection of light

[1]Department of Biology, Washington University in St. Louis, St. Louis, MO 63130, USA. [2]Washington University in St. Louis Center for Cellular Imaging, St. Louis, MO 63130, USA. [3]Present address: Bayer Crop Sciences, Chesterfield, MO, USA. [4]Present address: Roche Pharma Research and Early Development, F. Hoffmann-La Roche Ltd, Basel, Grenzacherstrasse 124, 4070, Switzerland. ✉e-mail: rdvierstra@wustl.edu

intensity, duration, direction, and spectral character. In most cases, Pr is the dark-adapted, inactive conformer while Pfr is the active conformer produced from Pr by red-light excitation, but there are unique variants where Pfr not Pr serves as the dark-adapted state thus requiring far-red light for photoactivation (bathyPhys), and divergent relatives that use other wavelengths for photointerconversion that collectively span the near UV to near infrared spectrum[1-5]. Phys will also thermally revert from Pfr back to Pr. This process primarily attenuates long-term signaling, but in some situations facilitates thermal sensation through accelerated depletion of the Pfr pool as temperatures rise[6,7].

Given the prominent roles of Phys in microbial life cycles and numerous agronomically essential processes in plants[4,8], and their engineering as manipulatable optogenetic reagents[3,9], defining their actions in molecular detail is of considerable interest. Unfortunately, while our understandings of downstream events are emerging in several phyla[3,4,8,10], it remains unclear how Phys directly translate light into a conformational change that drives photoperception.

Phys typically exist as homodimers bearing an N-terminal photosensory module (PSM) assembled from sequential Per/ARNT/Sim (PAS), cGMP-specific phosphodiesterase/adenylyl cyclase/FhlA (GAF), and Phy-specific (PHY) domains, followed by a C-terminal region important for dimerization and potentially signaling[1,3,11]. The bilin is buried within a GAF domain pocket, which through an extensive array of chromophore/protein interactions generates its unique absorption and photochromicity[12,13]. From structural studies with the PSM from several bacterial versions designated bacterial Phy photoreceptors (BphPs), including that from *Deinococcus radiodurans* (*Dr*BphP)[14,15], rudimentary understandings of the initial steps after light excitation have emerged. Key and possibly universal events include reorientation of the covalently bound bilin from a *ZZZssa* to *ZZEssa* configuration through *cis*-to-*trans* isomerization at the C15 = C16 methine bridge, which flips the amphipathic D pyrrole ring and promotes sliding of the bilin within the GAF domain pocket[16-18]. This movement forces the PHY domain hairpin (or tongue) loop that contacts the GAF domain, to refold from β-stranded to α-helical, whose specific and subsequent rebinding to the GAF domain rotates the PHY domain while shortening the distance between the GAF and PHY domains (at least in PSM-only models)[14,15]. This "pull" presumably instigates additional conformational change(s) that allosterically generate a signaling competent Pfr state.

While plant Phys employ a unique head-to-tail dimeric PSM architecture to signal through Pfr-specific binding to a collection of downstream effectors[19,20], many microbial BphPs associate their PSMs head-to-head, which often connect to paired output modules (OPMs) through coiled-coil regions (defined here as signaling (S) helices)[21][14,15,22-27]. Besides promoting dimerization, the BphP OPMs contain one of a number of output motifs, the most common being a transmitter (or two-component) histidine kinase (HKs) bidomain that initiates a phosphotransfer signaling cascade[1,3,11]. Here, the catalytic ATPase (CA) domain within the HK bidomain directs phosphorylation of a conserved histidine in an adjacent dimerization histidine phosphotransferase (DHp) domain in *cis* or *trans*, followed by transfer of the bound phosphate to a conserved aspartate in an often separate response regulator domain that further directs signaling[21]. Other output motifs are also prevalent within the BphP family, including diguanylyl cyclase, diguanylate phosphodiesterase, methyl-accepting, phosphatase 2 C, and two-helix output sensor, thus illustrating a wide range of signaling outcomes. Based on several examples[22-25,27], it is thought that the PSM transfers the light signal to changes in downstream OPM activity through allosteric modification of the S-helix connections. Possible mechanisms include splaying, rotation, and/or changes in register of the paired S-helices as BphPs photoconvert from their resting to photoactivated states.

Part of the challenge in fully understanding how microbial BphPs signal has been the lack of sufficiently resolved 3D structures of the full-length dimers in both their Pr and Pfr states, presumably due to mobility of the OPMs. Here, we successfully developed models of both states by cryo-electron microscopy (EM) using a BphP with HK activity from the proteobacterium *Pseudomonas syringae*[19,20,28] to illustrate how this subtype might signal. While its physiological function(s) in *P. syringae* are unknown, its Pseudomonad relative *P. aeruginosa* uses an ortholog to regulate quorum sensing through light-dependent phosphorylation of its downstream target AlgB, a response regulator within the NtrC subfamily[10].

By cryo-EM of recombinant *Ps*BphP1 assembled with biliverdin (BV)[19,20,28], we generated 3D models of Pr and Pfr with the sharpest features seen for the PSM and S-helix regions (2.8–3.3 and 3.0–3.3-Å resolution, respectively). While the CA domains and part of the DHp domains were less resolved due to high mobility in both states, interpretable models were possible with lower resolution albeit more complete maps aided by RoseTTAFold and AlphaFold structural predictions of areas interpretable at the secondary and tertiary structural levels. From this ensemble of models, we identified how Pr→Pfr photoisomerization of BV translates from the PSMs into the paired HK regions through hairpin refolding followed by displacement of the PHY domains and scissoring of the adjacent S-helices connecting the PSMs to the HK bidomains. These collective motions dislodge the CA domains from their DHp domain-bound inhibitory positions to enable autokinase-competent structural states where the ATP-binding pockets move closer to the phosphoacceptor histidines. Phosphotransfer assays confirmed our prediction that *Ps*BphP1 effectively modifies *Ps*AlgB via a canonical transmitter kinase reaction, but surprisingly without forming a tight complex either as Pr or Pfr. Taken together, these BphP models illustrate how light-driven reconfiguration of the bilin allosterically translates long range into altered kinase activity by this subclass of Phy photoreceptors.

## Results

### Expression of *Ps*BphP1 as a full-length dimer assembled with biliverdin (BV)

Full-length *P. syringae* BphP1 (Fig. 1a) assembled covalently with BV and modified with an N-terminal 6His tag followed by a tobacco etch virus (TEV) protease cleavage sequence, was synthesized by simultaneous expression of the tagged polypeptide with the *Synechocystis* heme oxygenase (HO)−1 in *Escherichia coli* BL21(DE3) cultures supplemented with 5-aminolevulinic acid to boost BV synthesis[19,20,28]. Purification and removal of the tag by TEV protease yielded homogeneous preparations of photointerconvertible bili-protein as judged by SDS-PAGE and UV-vis absorption spectroscopy (Supplementary Fig. 1a, b). The final *Ps*BphP1 samples used here differed from wild type only by inclusion of a single N-terminal glycine remnant from the TEV protease recognition site.

Initial characterizations by cryo-EM revealed that *Ps*BphP1 preparations as Pr or Pfr tend to form aggregates in buffered solutions containing NaCl at various concentrations even in the presence of assorted detergents, additives, pH values, and/or cryo-EM grid treatments, which obfuscated high-resolution structural studies. Fortunately, a screen of various buffers revealed that moderate concentrations of KSCN produced monodispersed *Ps*BphP1 samples. Analysis of preparations in 175 mM KSCN indicated normal Pr/Pfr absorption, an indistinguishable spectral change ratio for the Pr/Pfr difference spectra, and near equivalent Pfr→Pr thermal reversion rates (Supplementary Fig. 1b, c). Measures of autokinase activity using [δ-$^{32}$P]-ATP[19] showed faster Pfr-dependent kinetics with *Ps*BphP1 samples in KSCN than those in NaCl, indicating that KSCN increases the activity of the output module (Supplementary Fig. 1d). Negative stain EM also showed a similar ensemble of 2D particle images compared to samples dissolved in solutions containing NaCl (Supplementary Fig. 2). From these results, we found that *Ps*BphP1 retains strong HK activity and near native photocycles in the presence

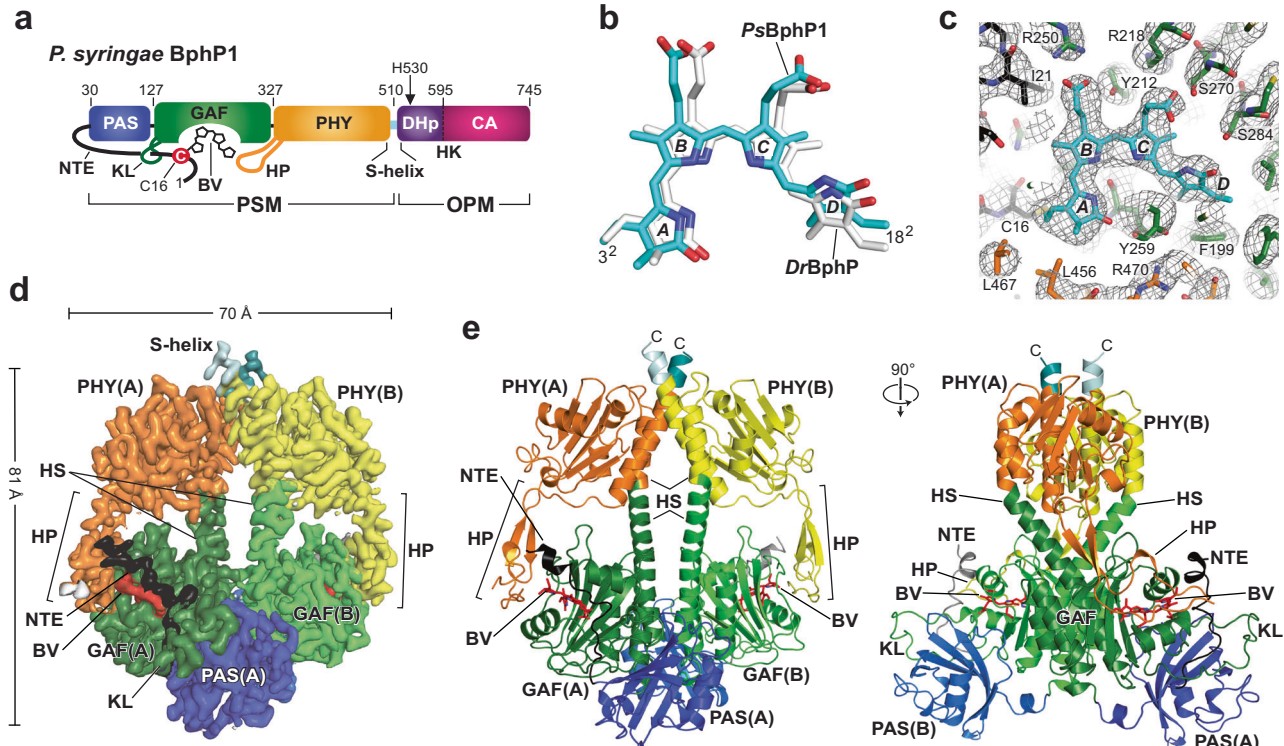

**Fig. 1 | High resolution 3D structure of the PSM from the *P. syringae* (*Ps*) BphP1 dimer as Pr. a** Domain organization of *Ps*BphP1. Shown are the positions of BV, the PAS, GAF, and PHY domains, and the NTE, knot lasso (KL) and hairpin (HP) features, portions of the S-helix linker, and the DHp and CA domains within the HK bidomain of the OPM. The numbers delineate domain boundaries within the polypeptide. Cys16, which forms a thioether linkage with BV is identified by the red circle. His530, which provides the phosphoacceptor site in the DHp, is indicated by the arrow. The length of the NTE was extended for clarity. **b** Relative positions of BV as Pr in the *Ps*BphP1 dimer (PDB 8U4X) with that determined previously by X-ray crystallography of a PSM fragment from *D. radiodurans* (*Dr*) BphP (PDB 4Q0J)[29]. The nitrogen, oxygen, and sulfur atoms are in blue, red, and yellow, respectively. The A-D pyrrole rings are labeled. The $3^2$ carbon that forms the thioether linkage with the apoprotein and the $18^2$ vinyl carbon are labeled for reference. **c** Cryo-EM

model of the BV-binding pocket from the A protomer (sticks) of *Ps*BphP1 superposed with the EM map (gray mesh). BV and the NTE, GAF domain, and PHY hairpin carbon atoms are in cyan, black, green, and orange, respectively. **d** Surface-rendered 2.8-Å resolution view of the 3D EM map of the dimeric *Ps*BphP1 PSM. EM density was rendered at 5 σ. EM density within 3 Å of a modeled atom was color-coded by domain with NTE, PAS, GAF, PHY, S helix, and BV colored in black, blue, green, yellow/orange, teal/light cyan, and red, respectively; those at greater distances were rendered white. Positions of the helical spines (HS), HP, NTE, and KL features are highlighted. Dimensions of the dimeric PSM are indicated where the width excludes NTE and hairpin contributions. **e** Orthogonal cartoon views of the dimeric PSM model of *Ps*BphP1 as Pr. Features are colored as in panel (**d**). BV is shown in red sticks. C, C-terminus.

of KSCN, thus leading us to conclude that KSCN at the concentrations used did not appreciably disrupt the native 3D structure of *Ps*BphP1 dimers and thus was suitable for cryo-EM.

## Cryo-EM models of *Ps*BphP1 as Pr

From *Ps*BphP1 preparations dissolved in a HEPES-NaOH (pH 7.5) buffer containing 175 mM KSCN, we generated cryo-EM maps of Pr and Pfr using either dark-adapted samples containing solely Pr or samples irradiated to steady state levels of Pfr (~80% of total Phy pool) with red light immediately before freezing. As shown by the work flows of both samples, we collected a set of highly resolved 2D class averages for both Pr and Pfr which enabled construction of informative 3D maps (Supplementary Figs. 3–5). From 3,233,515 picked particles, 1,151,696 were sufficiently similar to produce high quality 2D class averages for Pr. The combined averages then generated an EM density map of the PSM at 2.9-Å resolution which was further improved to 2.8-Å resolution through symmetry expansion and local refinement of 1,666,098 matching protomers (Supplementary Fig. 3b; Supplementary Table 1). The resulting data allowed construction of a head-to-head dimeric model encompassing residues 12-459 and 466-516 of protomer A and residues 11-459 and 466-516 of protomer B (Fig. 1d, e; Supplementary Table 1). While this 3D map provided high resolution for the PSMs, the HK bidomains and much of the S-helices were poorly visible probably due to high mobility.

The resulting high-resolution Pr model conformed well with prior PSM structures of BphPs[11,12,14,15,22,29–31] (Fig. 1d, e). Besides the covalent connection, the PAS and GAF domains were tightly linked non-covalently via hydrophobic interactions within the figure-of-eight knot created by a lasso loop extending from the GAF domain to tether the N-terminal extension (NTE) projecting from the PAS domain, which was further solidified by a noncovalent contact involving Ile27 at the knot center. BV was covalently linked via a thioether linkage to Cys16 within the NTE. As expected from other homodimeric Pr models, BV in both protomers assumed a *ZZZssa* configuration (Fig. 1b), and was fixed within the GAF domain pocket by an extensive network of conserved amino acid contacts to all four pyrrole rings and the B- and C-ring propionates (Fig. 1c; Supplementary Fig. 6a). Notable examples were the side-chain positions of: (i) Asp203 within the DIP motif and Tyr259 which move as a pair in hairpin binding as Pr and Pfr; (ii) His256 and the main chain carbonyl of Asp203 that help fix the pyrrole water at the center of the A-C pyrrole rings; (iii) Tyr172 and Phe199 near the D pyrrole ring that reorient during photoconversion; and (iv) Arg250 and Arg218 that bind the B-ring propionates as Pr and Pfr, respectively[12–14,22,23,31].

While most of these contacts are conserved among bacterial, cyanobacterial, and plant Phys[1], one notable exception in *Ps*BphP1 was Gly286. This residue is generally a histidine in canonical Phys, which hydrogen bonds with the D-ring carbonyl through its imidazole group

as Pr but then switches to provide an important anchor for the C-ring propionate in Pfr after the chromophore isomerizes[14,32,33] (Supplementary Fig. 6a). This substitution probably explains the slight positional difference of the bilin in the binding pocket by comparison to its *D. radiodurans* ortholog *Dr*BphP, with the absence of this histidine/propionate connection also possibly explaining the increased rate of Pfr→Pr thermal reversion for *Ps*BphP1 compared to that of *Dr*BphP (see Supplementary Fig. 1c and ref. 14).

The GAF and PHY domains in *Ps*BphP1 as Pr were linked covalently via the intervening helical spine and noncovalently through the hairpin extension from the PHY domain contacting the GAF domain near the chromophore (Fig. 1d). Its antiparallel structure was clearly β-stranded in both protomers as expected for Pr[14,15,22,25,26] (Supplementary Fig. 7a), and makes prominent contacts with the GAF domain using WGG and PRxSF motifs within the hairpin (WSG and PRTSF in *Ps*BphP1), and the DIP motif and an adjacent α-helix at residues 254-263 within the GAF domain (Fig. 1c; Supplementary Figs. 6a and 7a). As it exits the PAS/GAF bidomain, the NTE also assumed an α-helical fold that contacts the hairpin.

Improved resolution of the HK bidomain was then enabled by 3D variability analysis (3DVA), which isolated particle images with better congruity (Fig. 2; Supplementary Fig. 3c). These refinements culminated in a 3.3-Å resolution EM map that partially illuminated the HK bidomains, including an outline of the CA domain positions, and extended modeling of the sister DHp domain helices to Asn533 and Arg534 in protomers A and B, respectively, to now include the predicted phosphoacceptor histidines (His530) (Fig. 2a). Although the EM density for helix α2 was readily apparent, its signals became too diffuse for exact modeling shortly after residues 533/534, thus muddling the connecting turn between DHp helices α1 and α2 so that the amino acid registry was lost. Fortunately, map density for helix α2 then reemerged for 17−20 additional residues along with a rough outline of the mobile CA domains, which were placed on opposite sides of the DHp four-helix bundle and near the connecting turn between DHp helices α1 and α2 (Fig. 2c, d).

To better appreciate the relationship of the DHp and CA domains, especially with respect to the ATP-binding pocket in the CA domains and the phosphoacceptor histidine (His530) in the DHp domains, we generated a hypothetical structure based on the unsharpened volume map (Fig. 2b, c). Here, we overlaid the S-helix and DHp domain specified by AlphaFold for *Ps*BphP1 (https://alphafold.ebi.ac.uk/entry/Q885D3) onto our 3.3-Å resolution model, which allowed us to exploit this predicted model to position the connecting turn between DHp helices α1 and α2. Helix α2 placement was then adjusted manually into its EM density while retaining the relative amino acid registry of the two helices. Subsequently, we modeled the DHp four-helix bundle with left-handed helix α1 to α2 transitions and extended helices α2 as polyalanines to the end of the observable EM density for the DHp domains (Fig. 2d). Although the handedness of the turn was too ambiguous to model solely by the cryo-EM map, data described below for the Pfr state unambiguously identified the turn as left-handed. While not completely resolved, the short length of the DHp-CA domain linker and the resting positions of the CA domains along-side the DHp domains allowed predictions of the DHp(helix α2)-CA domain connectivity. Here, the CA domains associated non-covalently with the DHp domains via helix α1 of the DHp domain from its own protomer and helix α2 of the DHp domain from its sister protomer (Fig. 2c).

To help place the CA domain structure in the lower resolution model, we employed RoseTTAFold[34] to generate its predicted model ab initio. This domain model showed strong congruity with other CA or CA-like domains found in the PDB database as analyzed by the DALI Protein Structure Comparison Server[35]. Notably, one of the strongest matches was the 3D model for the ADP-bound CA domain determined empirically by X-ray crystallography from the *Thermotoga maritima*

HK853 transmitter histidine kinase (PDB 4JAV[36]) (Supplementary Fig. 8). As can be seen in Fig. 2c, the EM densities for both CA domains within *Ps*BphP1 as Pr, though weak, mimicked the overall shape of the predicted CA domain model when docked manually, with their expected placements near the DHp-CA domain transitions (Fig. 2c). Inspection of this assembly led us to conclude that the ATP γ-phosphate upon binding within the ATP-binding pockets as Pr would sit near the helix α1-α2 transitions at the top of the DHp domains, and thus reside ~30 Å away from the phosphoacceptor histidines (His530) (Fig. 2c; Supplementary Fig. 8). Given this distance, we predict that these positions would sequester ATP-binding and autophosphorylation from each other in *Ps*BphP1 as Pr, and thus dampen the kinase activity for this conformer as seen in vitro[19,20,28].

## Cryo-EM models of *Ps*BphP1 as Pfr

We then defined by cryo-EM a 3D model of the full-length *Ps*BphP1 dimer as Pfr by analyzing red light-irradiated samples. Here, 4,796,622 particle images were extracted and winnowed down to 1,029,141 after 2D classification, which revealed three similar but distinct EM density maps representing unique structural conformations of the photo-activated photoreceptor (Supplementary Figs. 4 and 5; Supplementary Table 1). Two maps resolved to 3.1- and 3.0-Å were dimeric head-to-head arrangements of the PSM with only short S-helix extensions visible beyond the PHY domains (Supplementary Fig. 9d–g). The PAS-GAF regions were intimately connected in the dimer like those in Pr while the paired PHY domains were either moderately or strongly splayed (designated "medial" and "splayed", respectively). Both the BV and hairpin conformations matched previous views of Phys as Pfr[14,15,22,25]. Much like the high-resolution Pr EM density map, these two Pfr maps lacked the HK bidomains and much of the S-helices presumably due to high mobility.

The third Pfr map resolved to 3.3 Å was most informative and revealed a surprising tetrameric arrangement whereby two full-length *Ps*BphP1 dimers abutted each other through their HK bidomains thus creating an elongated "dimer-of-dimers" (DoD) with the PSMs located at opposite poles (Fig. 3a, b; Supplementary Fig. 5). This arrangement presumably limited the flexibility of the dimers to enable further definition of the S-helices to Ala526 in protomer A and Ala517 in protomer B just proximal to the phosphoacceptor His530 (Fig. 3c, d). This stabilized structure also permitted construction by focused refinement of a PSM EM density map to 3.0-Å resolution (Fig. 3g; Supplementary Table 1), and permitted reasonably accurate modeling of the DHp and CA domains for one of the protomers (Supplementary Fig. 10). Interestingly, detailed SEC analysis of *Ps*BphP1 in the KSCN buffer revealed that this DoD configuration arises from a dynamic equilibrium between dimeric and tetrameric forms. As shown in Supplementary Fig. 11, increasing concentrations of full-length chromoprotein as Pfr progressively assembled tetramers from dimers. For example, the mole fraction of tetramers was 0.52 at 0.117 mg mL$^{-1}$ and increased to 0.84 at 1.17 mg mL$^{-1}$.

Importantly, the PSM regions of all three maps (DoD, medial, and splayed) allowed unambiguous modeling of both protomers as Pfr as judged by sufficiently resolved *ZZEssa* conformations of the bilins that aligned well for all three maps except for slightly different tilts of the D rings (Supplementary Fig. 9c) and by the presence of obvious α-helical hairpins, which are both signatures of this spectral state (Fig. 3e, f; Supplementary Figs. 7a and 9d–g). Comparison of BV at high-resolution for the Pr model to that of the DoD Pfr model indicated that the *cis*-to-*trans* isomerization of the C15 = C16 methine bridge during photoconversion generated a 161° planar flip of the BV D ring relative to the GAF domain (Fig. 3e). BV also slid in the binding pocket to allow formation of a hydrogen bond between the D-ring pyrrole nitrogen and the Asp203 carboxylate group (Supplementary Fig. 6b). As mentioned above, *Ps*BphP1 harbors a glycine at residue 286 rather than the highly conserved histidine. As comparable His-to-Gly

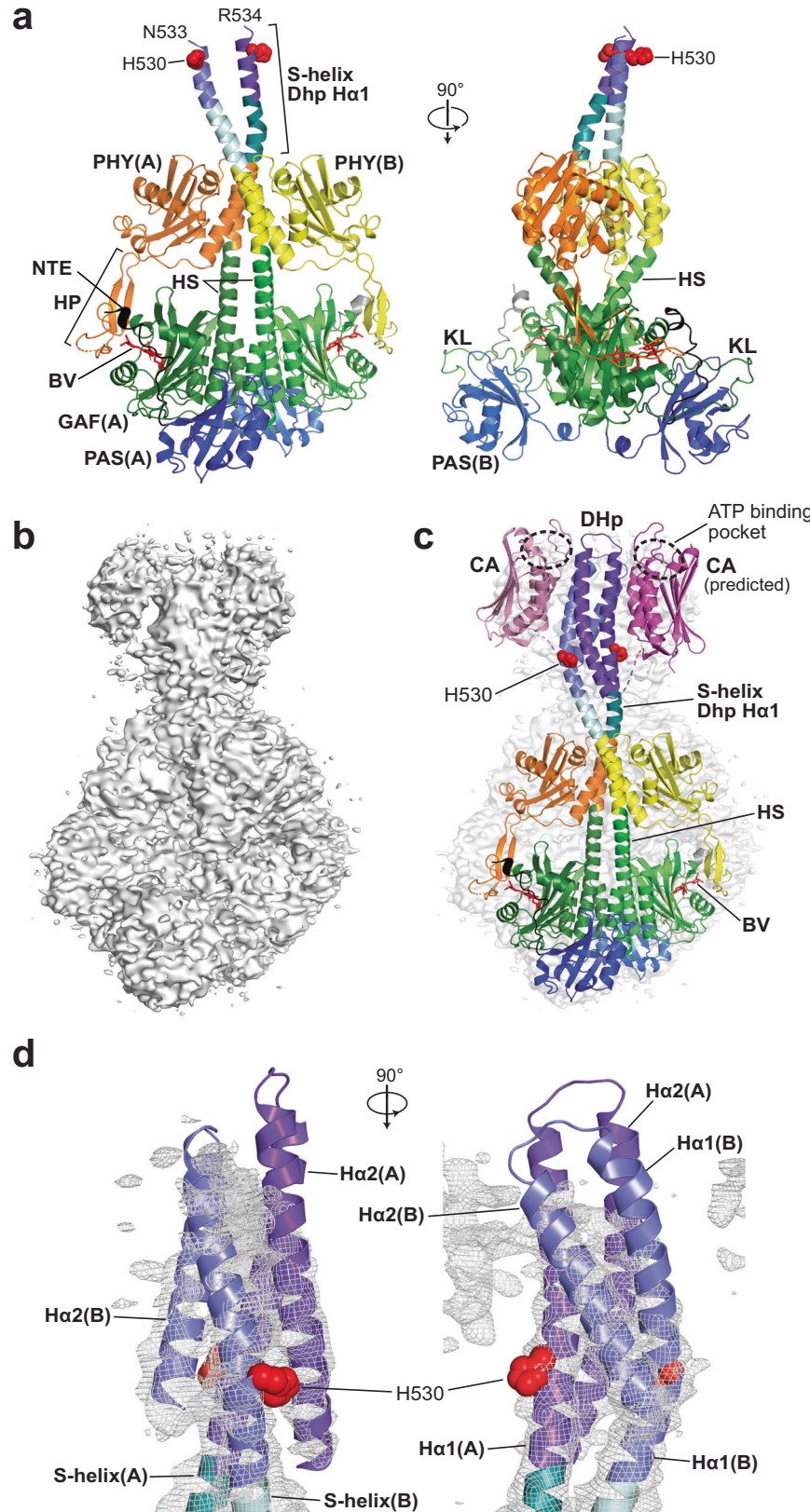

substitutions largely precludes photoconversion to Pfr in other Phys by removing a C-ring propionate contact in Pfr[32,33], we speculate that hydrogen bonding between Ser284 and the C-ring propionate compensates (Supplementary Fig. 6b). Related to this unusual connection, a serine is often found at this position in bathyPhys that use Pfr as the dark-adapted ground state[25,31], and has been shown to stabilize the Pfr

state indefinitely when introduced into the PSM of the plant PhyB isoform from *Arabidopsis thaliana*[19,32]. Another notable change in the GAF domain connections with the B-ring propionate was the use of Arg218 rather than Arg250 to anchor the propionate carboxylate group through a salt bridge (Supplementary Fig. 6b). In a similar vein, the hairpin modified its interactions after BV moved to its final Pfr

**Fig. 2 | Maps of *P. syringae* (*Ps*) BphP1 as Pr shown in low contour inform the relative positions of the S-helices and HK bidomains in the dimer. a** Orthogonal cartoon views of full-length *Ps*BphP1 as Pr (PDB 8U8Z) modeled from a 3.5-Å resolution map that allowed extension of the structure into the S-helix/DHp regions connecting the PSMs to the HK bidomains. Color scheme is the same as in Fig. 1d, with the purple coloring added for the DHp domains. His530 predicted to participate in the HK phosphorelay is shown as red spheres. **b** An unsharpened EM density map of full-length *Ps*BphP1 contoured at 2 σ reveals a more complete dimer. **c** Positioning of the protomers as cartoon models in the map from (**b**) showing the relative positions of the PSM and HK regions. The position of the loop between the

Hα1 and Hα2 regions of the DHp domains was extended from those in (**a**) using the predicted AlphaFold structure of *Ps*BphP1 (AF-Q885D3-F1). In the absence of sufficiently resolved CA domains, we docked a predicted cartoon model (shown in magenta) of the CA domain developed in RoseTTA-fold[34]. ATP-binding pocket in the CA domains is outlined by the dashed black ovals. **d** Orthogonal cartoon views of a predicted model for the DHp domains superposed with an unsharpened map of *Ps*BphP1 as Pr (gray mesh) contoured at 4 σ. The S-helix is colored in teal/light cyan and the DHp domains in purple. His530 is shown as red spheres. The CA domains were omitted for clarity.

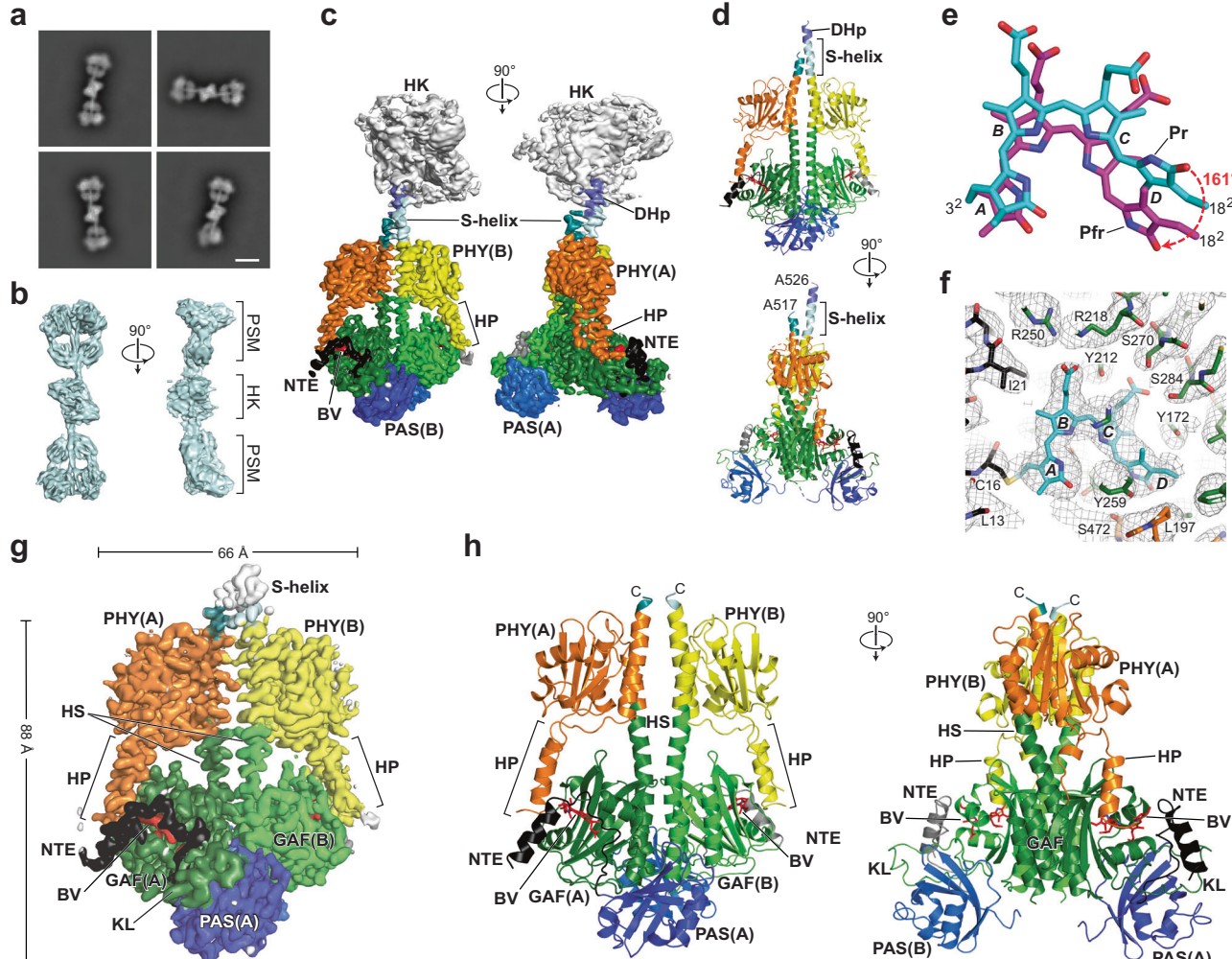

**Fig. 3 | 3D structure of the full-length *Ps*BphP1 dimer as Pfr based on the dimer-of-dimer (DoD) maps. a** 2D class averages generated from a collection of cryo-EM images showing the tetrameric assembly of *Ps*BphP1 into DoDs. **b** Orthogonal surface-rendered views of a 3D map reconstruction illustrating DoD assembly. C2 symmetry was applied to the map prior to symmetry expansion and local refinement (see Supplementary Fig. 5). **c** Orthogonal surface-rendered views of a 3D EM map of *Ps*BphP1 dimers as Pfr (PDB 8U62) that was generated by focused refinements of one of the PSMs and the DoD assembly point at the HK region (see **b**). The EM density was colored and labeled as described in Figs. 1d and 2a. **d** Orthogonal 3.3-Å resolution cartoon views of a dimer model deconstructed from the DoDs. Domains are colored as in (**c**). **e** BV positions after superposition of the GAF domains as Pfr (PDB 8U63) or as Pr (PDB 8U4X). The $3^2$ carbon that participates

in the thioether linkage with the apoprotein and the $18^2$ vinyl carbon are labeled for reference. Pyrrole ring D rotates ~161° after Pr→Pfr conversion. **f** Cryo-EM model of the BV-binding pocket as Pfr (PDB-8U63). The model was derived from the A protomer (sticks) and superposed with the EM map (gray mesh). BV and the NTE, GAF domain, and PHY hairpin carbon atoms are in cyan, black, green, and orange, respectively. **g** Surface-rendered view of the 3D EM map of the dimeric *Ps*BphP1 PSM as Pfr. The EM density was colored and labeled as in Fig. 1d. Dimensions of the dimeric PSM are indicated where the width excludes NTE and hairpin contributions. **h** Orthogonal 3.0-Å resolution cartoon views of the dimeric *Ps*BphP1 model as Pfr (PDB-8U63) generated with cryo-EM views encompassing just the PSM (see Supplementary Fig. 8c–f). The various features are colored as in (**g**). BV is shown in red sticks. Positions of the helical spines (HS), HP, NTE, and KL features are highlighted.

position with Ser472 in the hairpin PRxSF motif hydrogen bonding to Asp203 and Try259 in the GAF domain (Supplementary Fig. 6a, b).

As with the Pr data, resolution of the HK bidomains was possible by pointed scrutiny of the DoD map. Here, the HK regions were analyzed in isolation to produce a 4.1-Å map with adequate features to

define the overall shape of both DHp domains and the position of one of the two CA domains (Fig. 4; Supplementary Figs. 5 and 10). EM density for the second CA domain was too diffuse to detect, suggesting it was either not bound to the DHp domains or that its position in the tetramer was displaced by the sister dimer. These maps were then

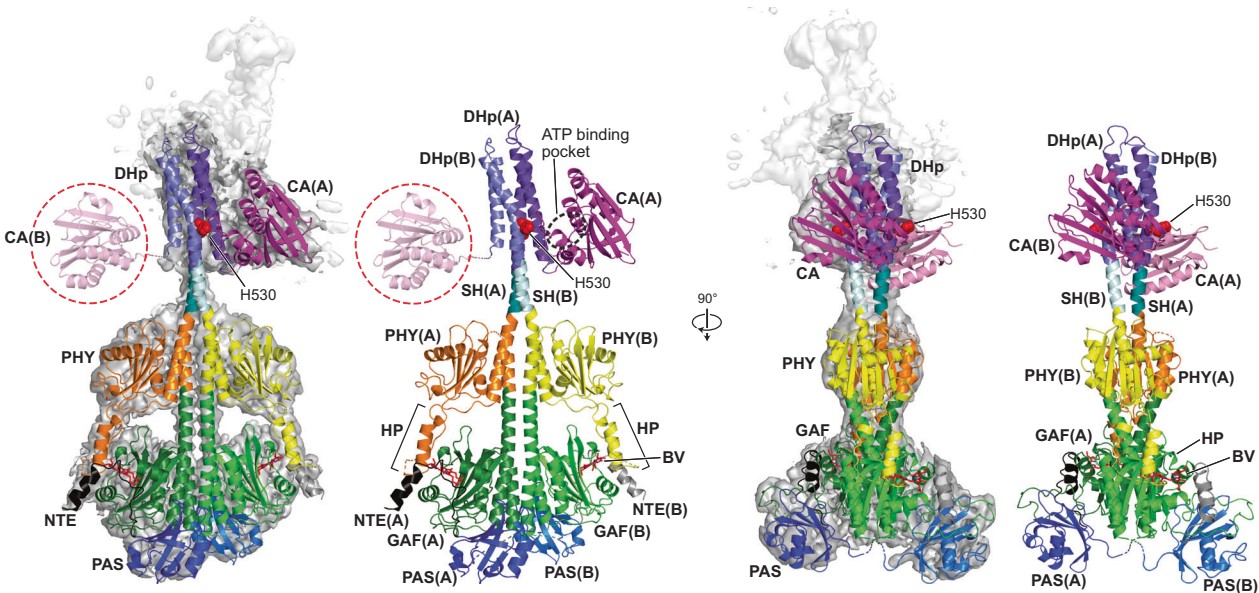

**Fig. 4 | Possible model of the *Ps*BphP1 dimer as Pfr based on cryo-EM maps of the DoD.** Shown are orthogonal views of the *Ps*BphP1 dimer either alone or superposed with a cryo-EM composite map of the region derived from the full-length DoD map and a focused refinement map of the DoD contacts at the HK bidomain (see Supplementary Figs. 5 and 10). The dark gray surface delineates one dimer, while EM density for the opposing dimer was rendered as a white surface.

Domains and features are colored as in Fig. 2. His530 is shown in red spheres. ATP-binding pocket of the CA domain in the A protomer is outlined with a dashed black circle. The dashed red circles locate the CA domain from the B protomer, which was added in an arbitrary orientation for completeness, but is absent in the EM density map. As seen in Supplementary Fig. 10, EM density dissipates at the terminus of helix α2 of the DHp domain of protomer B.

merged with the higher resolution DoD full-length map to produce a composite map of 4.4-Å, and from this map a full-length hypothetical model of one of the dimers from the DoD complex (Fig. 4). Here, modeling was conducted in the same manner as described above for Pr. In this case, we used the AlphaFold model of the DHp domain as a guide to manually extend its fit within the EM density. Placement of the observed CA domain into the density was then possible by manual rotation and translation of the entire CA domain predicted by RoseTTAFold, which oriented the CA domain near the bottom of the DHp and adjacent to the phosphoacceptor histidine.

Based on positioning of ADP within the CA domain of the *T. maritima* HK853 kinase bound with ADP (PDB 4JAV[36]), this orientation would place the ATP β-phosphate to within 7 Å of the phosphoacceptor histidine - His530 (See Fig. 4 and Supplementary Figs. 7 and 10). Consequently, we propose that photoconversion to Pfr dissociates both CA domains from their most favored positions in Pr to either become more mobile, or to relocate close to the bottom of the DHp domains. Either scenario would increase the local concentrations of histidine phosphoacceptors in the DHp domains and the catalytic ATP-binding sites in the CA domains to presumably accelerate ATP to histidine phosphotransfer.

**Possible mechanism for Pr→Pfr photoconversion**
Close inspection of an alignment of the Pr and Pfr models along with prior studies on other Phys[13–15,23,25–27,37,38], then enabled a possible detailed mechanism for *Ps*BphP1 photoconversion that eventually activates the autokinase activity of its HK bidomains (Fig. 5). After the light-induced *Za* to *Ea* flip of the D-pyrrole ring in BV, the PSM undergoes a cascade of conformational changes involving a number of features. As described above and from prior structures[14–18], the primary event caused by this flip is release and subsequent refolding of the hairpin from β-stranded to α-helical followed by rebinding of the hairpin to the GAF domain. These interactions are mostly hydrophobic through contacts involving residues 467-480 of the hairpin, which includes the entire α-helix, and the face of the GAF domain adjacent to the BV D-ring. Although hydrogen bonding is less represented, the

conserved bonds between the PRxSF motif Ser472 of the hairpin and Asp203 and Tyr259 of the GAF domain were observed. The hairpin transition was also coincident with rotation of the NTE helix by ~100°, which is stabilized in part by a salt bridge between Arg466 of the hairpin and Glu10 of the NTE. This bridge implies a potential role for the NTE in Pfr stabilization (Supplementary Figs. 6a and 7a).

As shown in Fig. 5b, d, e, the resulting hairpin reconfiguration has two apparent consequences. One is to "pull" the distal exit point for the hairpin closer to the GAF domain (Supplementary Fig. 7a). As an example, the Cα atoms of Gly177 and Ile483 were separated by 15.5 Å in Pr but only 9.5 Å in Pfr. Superposition either of an ensemble of 51 PHY domain 3D models currently available within the PDB database (Supplementary Fig. 12a, b), or only the four PSM structures described here for *Ps*BphP1 (i.e., one Pr and three Pfr models) (Supplementary Fig. 12c, d), revealed that the core PHY domain structure is remarkably rigid with little average structural displacement even among a wide range of Phy relatives with divergent sequences, photostates, and/or crystallization contacts. In fact, PHY domain flexibility was seen only at its entrance and exit points. Thus, any strain imposed on the PHY domains from the hairpin transitions should pass through the PHY domain core and impinge directly on the connecting GAF-PHY helical spine and PHY-OPM S-helices. The cumulative effects are to straighten the helical spine kink found in Pr following by torque on the S-helices, with the end result being a 35° rotation of the whole PHY domain (Fig. 5a, b, d, e).

The second consequence of the hairpin reconfiguration is to swivel of the GAF domain protomers relative to each other through a pivot point centered at Val307 (Fig. 5c). Our speculation is that this pivot buffers large motions at the PHY domain from dissociating the dimeric interface at the GAF domains, so that conformational energy can instead be directed toward the HK bidomains. Consequently, the PAS domains, whose function(s) have remained enigmatic, might provide a backstop to limit pivoting, while also supplying interdomain contacts to support PSM dimerization.

Finally, rotation of the PHY domains pulls outward the terminal α-helix of the PHY domain, which is continuous with the S-helix and helix

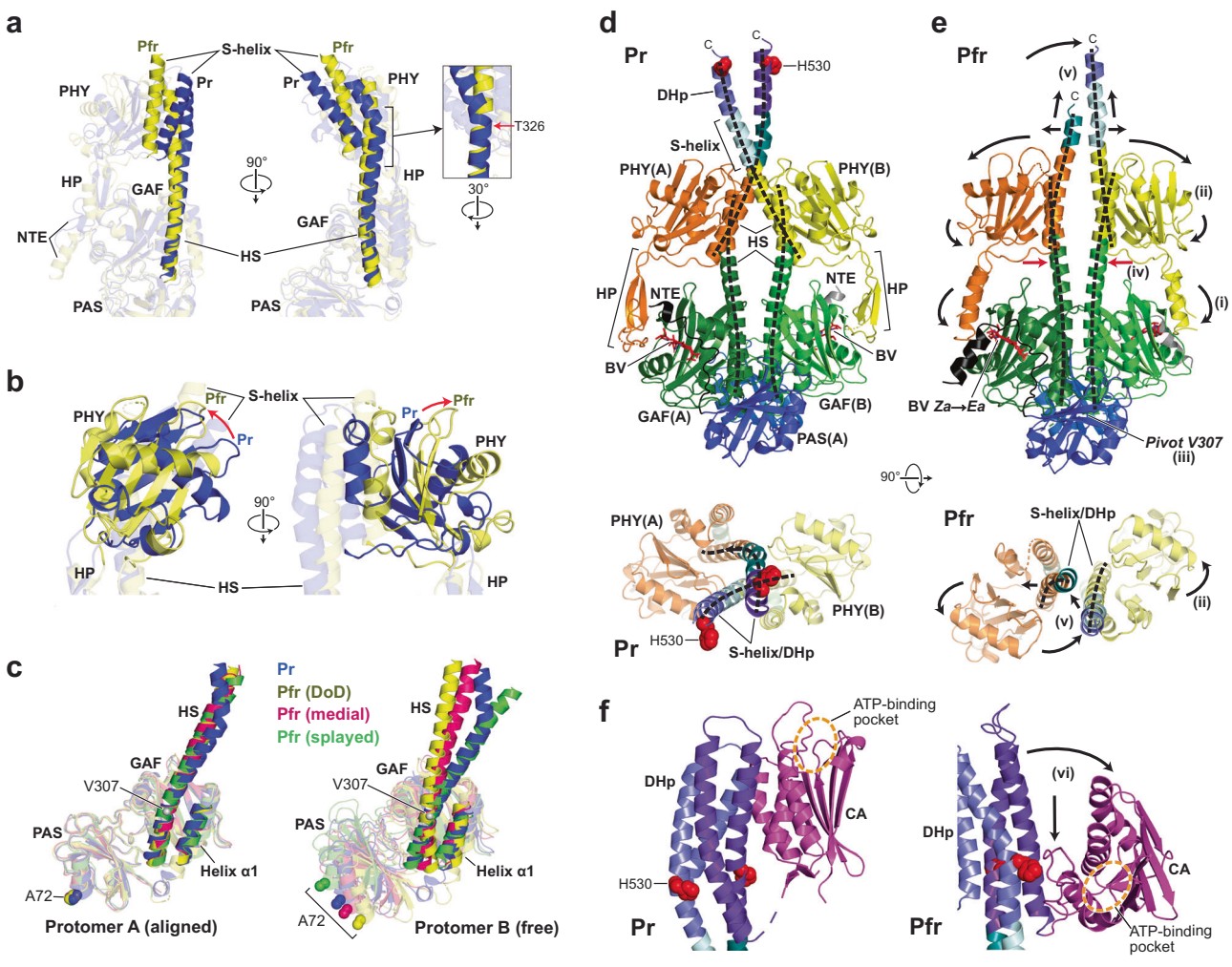

**Fig. 5 | Detailed conformational changes and a model for Pr→Pfr photo-conversion of full-length *Ps*BphP1. a** Straightening the GAF-PHY helical spine kink centered at Thr326 during photoconversion shown by orthogonal views of the Pr (blue) and Pfr (DoD) (yellow) cartoon models. **b** Rotation of the PHY domain enforced by the β-stranded to α-helical transition of the hairpin shown by ortho-gonal views of the Pr (blue) and Pfr(DoD) (yellow) cartoon models superposed via the GAF domains. **c** Light-induced pivot of the helical spine exiting the GAF domain around Val-307. GAF domains of protomer A were superposed for Pr, Pfr (DoD), Pfr (medial), and Pfr (splayed) models of *Ps*BphP1. (left) Cartoon models positioning the helical spine in protomer A. (right) Cartoon models highlighting the positions of the helical spine in protomers B after superposition of the GAF domain of pro-tomers A. The positions of Ala72 highlight the range of motion for the PAS-GAF

domain. **d** Orthogonal cartoon views of the dimer as Pr. Domains/features are colored as in Fig. 1d. **e** Orthogonal cartoon views of Pfr illustrating movements within the *Ps*BphP1 dimer upon photoconversion. Include are: (i) tugging of the hairpin closer to the GAF domain as the hairpin converts from an anti-parallel β-strand to α-helical, (ii) rotation of the PHY domains with coincident straightening of the helical spine, (iii) pivoting of the GAF domains around V307; (iv) movement of the GAF-PHY helical spines closer to each other (red arrow), and (v) scissoring of the sister S-helices. **f** Close-up view of the proposed positions of the CA domain as Pr and Pfr showing downward movement and rotation of the ATP-binding pocket closer to His530 (vi). The Pr and Pfr images were derived from the protomer A models described in Figs. 2c and 4. Dashed orange ovals identify the ATP-binding pocket.

α1 of the DHp, while straightening inward the GAF-PHY helical spine. As shown in Fig. 5d, e, such movements imbue a scissor motion to this α-helical feature in both protomers which converts the crossover arrangement of the paired S-helices in Pr to the roughly parallel or splayed arrangements in Pfr. As seen from a top view, the S-helix of protomer B in the DoD model undergoes a reasonably rigid and straight transition for the Pfr endstate, while the S-helix of protomer A straightens in Pfr and rotates around the S-helix in protomer B to allow intimate contact between the S-helices and DHp α1 helices (Fig. 5d, e).

From our partially resolved HK bidomains (Figs. 2c and 4), we hypothesize for *Ps*BphP1 and related BphPs with transmitter kinase activity (at least those with accelerated activity as Pfr) that this rear-rangement translates up into the DHp domains to dislodge the CA

domains from their Pr positions, thus allowing their ATP-binding pockets to interact more intimately with the phosphoacceptor His530 residue in the DHp domains as Pfr (Fig. 5f). This proximity should then enhance *Ps*BphP1 autophosphorylation as seen in vitro[19,20,28], and possibly improve access to the bound phosphate by the aspartate acceptor in its paired response regulator. The formation of DoDs is potentially telling about the mechanism of autophosphorylation. At concentrations where tetramers form (see Supplementary Fig. 10), the local concentrations of CA and DHp domains from a single dimer should be significantly higher. Consequently, if the CA domains had moderate affinity for the DHp domains at the histidine phos-phoacceptor position, the CA domains would easily outcompete neighboring dimers for DHp binding, which is not what was found. It is

apparent that a key aspect of the autokinase mechanism is that the CA and DHp domains only transiently contact each other as Pfr, which has special relevance to the results below.

### PsBphP1 interacts transiently with its response regulator AlgB

The substantial mobility of the HK bidomains seen with *Ps*BphP1, as well as within other microbial Phys[23,24] and transmitter HK bidomains more generally[21], led us and others to speculate whether its corresponding response regulator could stabilize this region in a ternary complex and thus might help full structural understanding of microbial Phys. In fact, ref. 23. recently attempted a short cut of this approach through cryo-EM analysis of full-length *D. radiodurans* (*Dr*) BphP ectopically fused to its dimeric phosphatase partner *Dr*BphR[28,39] via a short C-terminal linker. Here the PSM and parts of the S-helices and DHp region were well resolved, but the CA domains attached to *Dr*BphR were not, as also seen here for our high-resolution cryo-EM maps of *Ps*BphP1.

To avoid the inherent challenges that translational fusions present, we attempted to assemble a native *Ps*BphP1/response regulator complex without covalent coupling. At the time of this study, the phosphorylation target(s) of activated *Ps*BphP1 were not known even though it is a robust autokinase as Pfr[19,20]. No genetic links were available and its operon does not include possible candidates besides the HO for BV synthesis[28], unlike other bacterial Phys such as *D. radiorurans* BphP that include a response regulator in its photoreceptor operon[28,39]. Fortunately, a survey of other Pseudomonads by Mukherjee et al[10]. identified AlgB as likely candidate, which was originally discovered in *P. aeruginosa* as a NtrC-type response regulator whose phosphorylation by an orthologous BphP forms a central node integrating light to quorum sensing and biofilm behavior[10]. They also noticed that this BphP/AlgB signaling pair is widely distributed among proteobacteria (>150 species), including an ortholog in *P. syringae* (84% sequence identity for *Ps*AlgB; GenBank: MCF9017822.1).

To confirm that *P. syringae* AlgB works with *Ps*BphP1, we assayed for phosphotransfer using [δ-$^{32}$P]-ATP. Here, full-length *Ps*AlgB with its phosphoacceptor receiver (REC) domain followed by an AAA-ATPase domain (Fig. 6a), was expressed recombinantly with an N-terminal TEV-protease cleavable 6His tag, which was removed during purification to yield *Ps*AlgB containing a single N-terminal glycine remnant upstream of the initiator methionine (Fig. 6a). As shown in Fig. 6b, c, 2 μM of *Ps*BphP1 preloaded with phosphate as Pfr, readily transferred the phosphate to *Ps*AlgB (1 μM) with an apparent rate constant of 2.8($\pm$0.3 SE)10$^{-1}$s$^{-1}$ at 22 °C. The reaction was essentially completed within 10 min while the initial Pfr-specific loading of *Ps*BphP1 required ~1 h to saturate (see Supplementary Fig. 1d; ref. 19). As expected[10], alanine substitution mutants showed that the initial autophosphorylation of *Ps*BphP1 required the predicted internal phosphoacceptor His530, while subsequent phosphotransfer to *Ps*AlgB required the predicted phosphoreceiver Asp59 in *Ps*AlgB (Fig. 6d). In the latter case, the Asp59-Ala mutant of *Ps*AlgB failed to accept $^{32}$P even in the presence of highly labeled, wild-type *Ps*BphP1.

We then tested for assembly of *Ps*AlgB/*Ps*BphP1 complexes by SEC, in this case using the BphR response regulator/phosphatase from *D. radiodurans* as a control. Unlike *Ps*AlgB, *Dr*BphR is composed on just the REC domain (Supplementary Fig. 13a) but was previously found to be a phosphorylated by *Ps*BphP1 at least in vitro[28]. As shown in Fig. 6e, *Ps*BphP1 tested at a concentration that would mostly assemble DoDs as Pfr (see Supplementary Fig. 11c) but remain dimeric as Pr, failed to form a stable complex with *Ps*AlgB either as Pr or Pfr as assayed by the lack of an SEC elution shift for either component (Fig. 6e). *Dr*BphR also did not show a mobility shift by SEC when mixed with either of the two *Ps*BphP1 conformers (Supplementary Fig. 13b).

Unexpectedly, when *Ps*BphP1 and *Ps*AlgB were mixed in the presence of 1.5 mM ATP, detectable binding was also absent but robust hydrolysis of ATP was discovered, which was seen as a shift in the elution position of ATP to that of ADP. This shift was barely detected when either *Ps*BphP1 or *Ps*AlgB were incubated alone with ATP (Fig. 6f), but occurred when mixing *Ps*AlgB with either the Pr or Pfr conformers of *Ps*BphP1 (Fig. 6g, h), indicating that the two proteins together elicited robust ATPase activity for *Ps*AlgB. For a control, neither the Pr nor Pfr states of *Ps*BphP1 showed this ATP hydrolysis when *Dr*BphR was added instead (Supplementary Fig. 13b). We presumed that the strong and rapid ATPase activity stimulated by *Ps*AlgB was caused by an intrinsic ATPase activity in *Ps*AlgB given that both *Ps*BphP1 and *Ps*AlgB remained phosphorylated for hours when incubated together in the phosphotransfer assays (Fig. 6b). This activity might be a common feature of NtrC-type response regulators harboring an AAA-ATPase domain (Fig. 6a). In sum, it appears that *Ps*BphP1 and *Ps*AlgB work together in a transmitter kinase phosphorelay but through a transient interaction. Taken further, this transience implies that the internal interactions between the DHp and CA domains must be equally ephemeral at least after phosphorylation to ensure that *Ps*AlgB can bind the phosphohistidine intermediate.

## Discussion

Here, we present near complete views of a native Phy transmitter kinase in its dark-adapted and photoactivated states to reveal with sufficient resolution how the most common microbial Phy type signals as a head-to-head dimer via a transmitter phosphotransferase cascade. When the paired models of *P. syringae* BphP are combined with prior models of *Idiomarina* sp. A28L PadC[22,37] and *Xanthomonas campestris* BphP[25] that act either as a light-regulated diguanylyl cyclase or presumably through photoregulated protein-protein interactions, respectively, a prototypic coupling mechanism emerges for bacterial Phy signaling that exploits a swivel of flexible S-helices to connect light absorption to effector signaling.

While much of the PSM and S-helices for *Ps*BphP1 were visualized here to high resolution, models for the HK bidomains were assembled with lower resolution maps augmented with structural predictions using RoseTTAFold and AlphaFold, and manual placement of the predicted CA domains into the maps. The Pfr model was also enabled by an unexpected assembly of *Ps*BphP1 into tetrameric DoDs at high concentrations, whose HK bidomain maps could be deconvoluted for reasonably accurate CA domain positioning for one of the protomers. That the DoDs retained normal Pr/Pfr photointerconversion and phosphotransferase activity implied that their formation did not structurally impair Pfr, and, in fact, might have helped identify novel features of the highly flexible HK bidomains (see below).

Comparisons of the high-resolution Pr and Pfr structures revealed a complicated gymnastics cascade that drive *Ps*BphP1 photoconversion (Fig. 5d–f), some of which have been seen previously with other BphPs and thus likely relevant to most members of the Phy superfamily[17,18,22–27]. The transition begins as expected with *ZZZssa*→*ZZEssa* photoisomerization and reorientation of the bilin in the GAF domain pocket followed by a complex sequence of rotations and conformational translations driven by the hairpin as it converts from anti-parallel β-stranded as Pr to α-helical as Pfr, which forces a downward rotational pivot of the PHY domain. The resulting prying force enabled by the structural stability of the PHY domain translates into motions of the connected helical spine (Fig. 5d, e), a feature which spans via several overlapping α-helices nearly the entire length of the photoreceptor, including the GAF, PHY, S-helix, and DHp features.

Ultimately, the paired S-helices extending downstream from the PSMs scissor, which adjusts the DHp interfaces from being asymmetric and out of register in Pr to roughly symmetric and in register in Pfr (Fig. 5d, e). Through a yet to be understood mechanism, this light-induced motion dislodges the CA domains from their primary resting positions against the DHp domains where the catalytic ATPase sites of the CA domains are widely separated from the phosphoacceptor histidine in the DHp domains (~30 Å), to ones where the γ-phosphate of

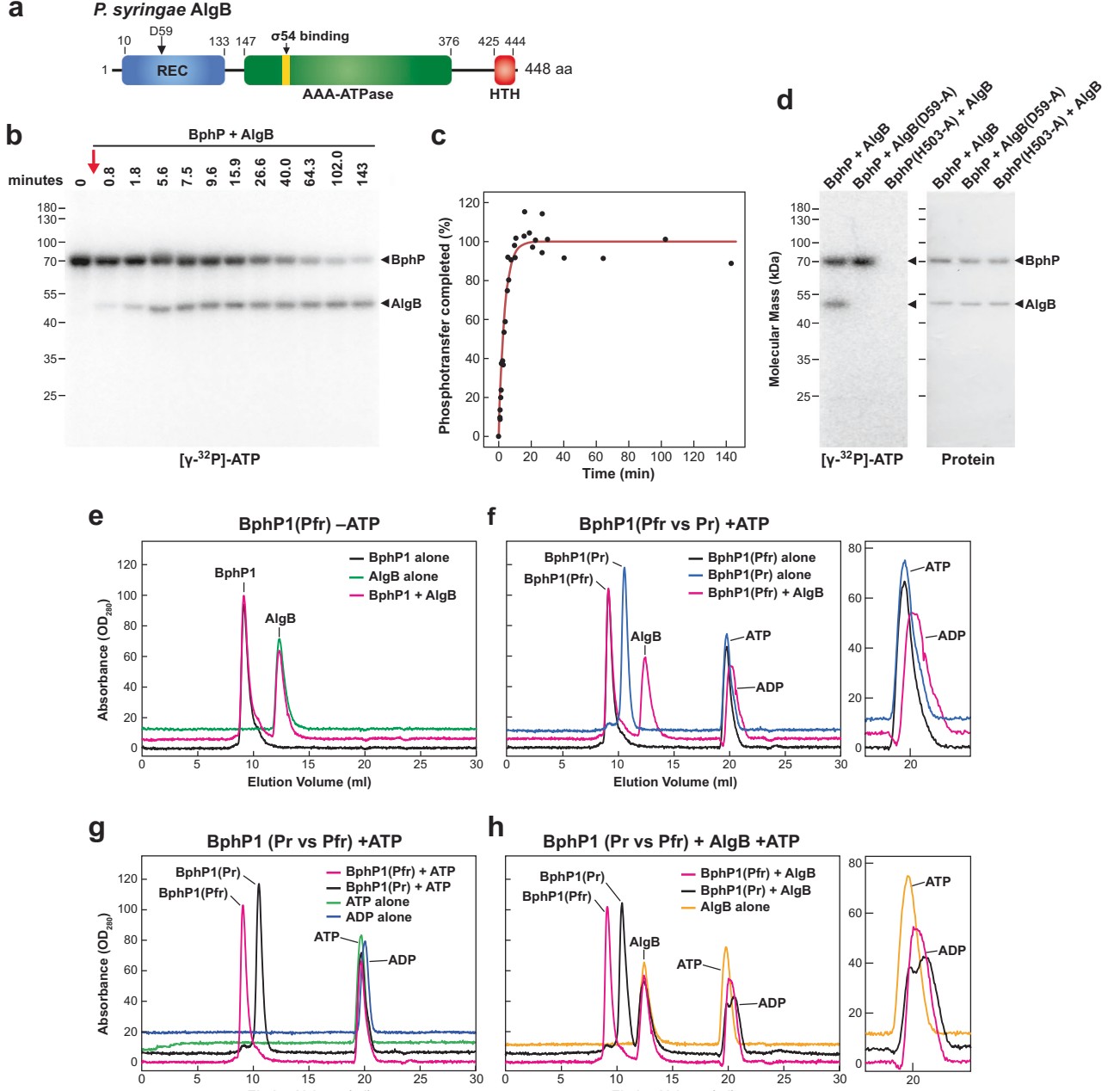

**Fig. 6 | The *P. syringae* AlgB works downstream of *Ps*BphP1 in phosphotransfer but without a tight interaction. a** Domain architecture of *Ps*AlgB. Asp59, predicted phosphoacceptor site by *Ps*BphP1[10]. REC, phosphoacceptor receiver domain. AAA, ATPase associated with various activities. HTH, helix-turn-helix DNA-binding motif. **b**, **c** Phosphotransfer kinetics from *Ps*BphP1 to *Ps*AlgB. *Ps*BphP1 was prelabelled as Pfr with [γ−32P]-ATP for 2 h and at *t* = 0 incubated further with *Ps*AlgB. Samples were subjected to SDS-PAGE and autoradiography (**b**) and quantified for 32P transfer by densitometric scans of the autoradiograms (**c**). Data in (**c**) represents three separate reactions. **d** Phosphotransfer from *Ps*BphP1 to *Ps*AlgB works via a canonical transmitter HK mechanism. Wild-type *Ps*BphP1 or its His530-Ala mutant were incubated for 2 h as Pfr with [γ−32P]-ATP, and then mixed with a twofold molar excess of either wild type *Ps*AlgB or its Asp59-Ala mutant. Reactions were subjected to SDS-PAGE and autoradiography, followed by staining for protein with Coomassie Blue. *Ps*BphP1 and *Ps*AlgB are indicated by the arrowheads. **e**–**h** SEC chromatograms of *Ps*BphP1 as Pr or Pfr with or without *Ps*AlgB and/or 1.5 mM ATP. *Ps*BphP1 at the concentration used (3 mg mL⁻¹) assembles as a dimer as Pr but mostly as a DoD as Pfr (see Supplementary Fig. 11). *Ps*AlgB was added at a concentration of 3 mg mL⁻¹. **e** Profiles for *Ps*BphP1 as Pfr and *Ps*AlgB either alone or mixed without ATP. **f** Profiles of *Ps*BphP1 as Pr or Pfr either alone or mixed with *Ps*AlgB and ATP. **g** Profiles of *Ps*BphP1 as Pr or Pfr alone with ATP. The elution profiles of ATP and ADP are included for comparison. **h** Profiles of *Ps*BphP1 as Pr or Pfr in the presence of both *Ps*AlgB and ATP. Elution profiles of ATP and ADP alone, or ATP mixed with *Ps*AlgB alone are included for comparison. A closeup of the elution region for ATP is included in (**f**) and (**h**).

ATP is now predicted to be only 7-Å away. As seen by kinase activity assays, this displacement in part generates a Pfr state with robust autophosphorylation and phosphotranferase activities (at least in vitro[19,28]; this report).

As an advantage of cryo-EM, we were able to describe four major structural forms for *Ps*BphP1, one as Pr and three as Pfr, which provided insight into how a Phy two-component HK sensor might work. As Pr, the PSM was relatively rigid, but had sufficient S-helix flexibility to

blur the overall picture of HK bidomain. Fortunately, some populations were similar enough to discern the relative positions of the DHp and CA domains and allow a plausible prediction for autokinase inhibition via the spatial separation of reactants. The Pfr maps presented an even more complicated picture where the three divergent conformational states seen here (DoD, medial, and splayed) implied that the S-helices become more pliable and possibly even allow dissociation of the HK bidomains as inferred by the conformation of the "splayed" Pfr model. Although the data for the DoD form provided structural appreciations for positioning of the HK bidomains, similar data for the splayed and medial forms of Pfr were absent, suggesting that the HK bidomain positioning is highly dynamic in these states. Moreover, it was evident that transitions between the four structural states included variations in the observed dimeric contacts at the PAS and GAF domains, which we propose helps the Phy dimer maintain structural integrity during photoconversion without complete dissociation.

The scissoring motions of the S-helices between the Pr and Pfr states are analogous to those proposed for a tandem pair of BphPs from *Rhodopseudomonas plaustris*[27], and those seen as Pr and Pfr for PadC from *Idiomarina sp. A28L* that terminates in a diquanylyl cyclase domain[22] and for *Xanthomonas campestris* (*Xc*) BphP bearing a PAS9-binding domain OPM[25]. Despite analysis of a Pr/Pfr heterodimer or a bathyPhy that prefers Pfr as the dark-adapted state, respectively, *Id*PadC and *Xc*BphP displayed strong photostate driven rotations of the PHY domain that disrupt the positions of S-helical linkers that join their PSMs to their disparate OPMs. While the OPMs in *Id*PadC swivel as Pfr while remaining in contact, those for *Xc*BphP completely splay, likely driven by weaker interactions between the C-terminal PAS9 domains. We saw little change in S-helix register for *Ps*BphP1 as proposed for asymmetric activation of *Id*BphP[22], but did see a substantial change in registry for the downstream DHp domains, which were slightly offset in the Pr state, but roughly symmetric as Pfr. The recently described paired Pr/Pfr models of full-length *D. radiodurans* BphP translationally fused to its cognate phosphatase *Dr*BphR through a short linker also emphasized roles for the paired S-helices during photoconversion[23]. In this case, Pfr formation was proposed to unzip their dimeric association to further transmit conformational flexibility into the OPM, but presently it is unclear how well this artificial construction reflects the natural situation. In fact, other than the β-stranded to α-helical reconfiguration of the hairpin, the Pr and Pfr models of the *Dr*BphP-*Dr*BphR fusion[23] showed little disagreement in structure when superposed as opposed to the more widespread changes seen here for native *Ps*BphP1. Furthermore, it remains ambiguous from this fusion model how the tethered *Dr*BphR moieties could be sufficiently displaced to reach the phosphoacceptor histidine upon *Dr*BphP photoconversion.

More universally, the signaling mechanism seen in detail for *Ps*BphP1 is remarkably similar to other types of transmitter kinases that assume a winged HK-bidomain architecture for the dimeric OPMs combined with a pair of short intervening α-helices that translate motional information from the N-terminal sensory domain[21]. Microbial Phys have simply adapted this paradigm to light signaling by incorporating a bistable light-sensitive sensory feature.

We propose that the sequence of photoconversion events also helps explain several peculiarities of Phys. One is the need for the N-terminal PAS domain and its subtending figure-of-eight knot, which are found in most, if not all, canonical Phys/BphPs. Together with their dimerization contacts, the PAS/knot forms a tight tether to the GAF domain[1,3,11], which we speculate provides a backstop to direct motions generated first by chromophore and then by the hairpin toward the PHY domains, with the pivot point centering at V307 in the GAF domains. The second peculiarity is the role(s) of the PHY domain and its extending hairpin. We and others proposed based on X-ray crystallographic models of the just the PSM that the "pull" generated by the

hairpin as it reconfigures from β-stranded to α-helical would splay the PHY domains and by extension, dissociate the S-helices and OPMs[14,15]. A similar uncoupling in the OPM was also proposed for the *Dr*BphP-*Dr*BphR ectopic fusion as Pfr[23]. However, our more complete model of *Ps*BphP1 together with the activated model for *Idiomarina* PadC[22,25] show that PHY domains remain in contact but generate a scissor rotation of the S-helices that translates into the DHp region. Our structural comparisons of a wide array of Phys with divergent amino acid sequences and spectral properties, revealed that the PHY domain is extremely rigid, which we presume is necessary to efficiently direct the prying force generated by the hairpin contraction toward the helical spine and into the S-helices as opposed to actuating PHY domain unfolding.

We note that binding of *Ps*BphP1 to its cognate response regulator *Ps*AlgB is quite transient in nature with no binding detected by SEC. Notwithstanding the lack of coelution, phosphotransfer from *Ps*BphP1 to *Ps*AlgB is rapid in vitro, and requires as predicted the phosphoreceiver Asp59 residue in *Ps*AlgB along with phosphoacceptor His530 on *Ps*BphP1. This is reminiscent of the fleeting interaction between the bacterial Phy Agp1 of *Agrobacterium fabrum* and its cognate response regulator *At*RR1[39], and might be required for efficient signal transfer in vivo. Unfortunately, the unstable nature of the *Ps*BphP1/*Ps*AlgB interaction likely negates observing this association via cryo-EM and possibly even X-ray crystallography and might necessitate alternative approaches to generate sufficiently high local concentrations of the pair for observation. While the physiological function(s) of the *Ps*BphP1/*Ps*AlgB transmitter kinase system are currently unknown, its orthology to that in *P. aeruginosa* suggests an evolutionarily ancient signaling mechanism that connects quorum sensing and biofilm formation to the light environment of this devastating plant pathogen.

## Methods

### Protein expression and purification

The coding region for the full-length *Ps*BphP1 polypeptide (residues 1-745[28]) was placed by polymerase-incomplete primer extension[40] into a modified pBAD vector in-frame with that encoding an N-terminal 6His tag followed by a TEV protease cleavage site (MGSSHHHHHHSSENLYFQG)[19]. This vector was then used to produce a plasmid containing *Ps*BphP1 with the His530-Ala mutation via Gibson assembly[41]. Coding sequences of these plasmids and all others used in this study were verified as correct by full DNA sequence analysis. 6his-TEV-*Ps*BphP1 and 6his-TEV-*Ps*BphP1(H530A) under the control of the *araBp* promoter were co-expressed in *Escherichia coli* BL21 DE3 cells with *Synechocystis sp.* PCC 6803 HO1 transcribed under control of the *lac* operon in the pET24a plasmid[28]. The cultures were grown at 37 °C in terrific broth to an $OD_{600}$ of 1. Heme synthesis was then enhanced by adding 5-aminolevulinic acid (5-ALA) to ~40 mg/L and the incubation temperature was reduced to 24 °C. One hour after 5-ALA addition, HO1 expression was induced by adding isopropyl β-D-1-thiogalactopyranoside (IPTG) to 1 mM and the culture incubation temperature was further lowered to 16 °C. One hour after IPTG addition, L-(+)-arabinose was added to a final concentration of 2 g/L to induce *Ps*BphP1 polypeptide synthesis. After incubating the cultures for 16 h in darkness, the cells were harvested by centrifugation, frozen under dim green safelights directly in liquid nitrogen, and stored at −80 °C.

*Ps*BphP1 (wild type and mutant) was purified in darkness or under dim green safelights at 4 °C or wet ice temperatures. Frozen cells concentrated to 300 mL from 6 L of cultures were disrupted by sonication on ice immediately after addition of 600 mL of Ni-nitrilotriacetic acid (Ni-NTA) column-binding buffer (10% (v/v) glycerol, 0.05% (v/v) Tween-20, 500 mM NaCl, 15 mM imidazole, 2 mM 2-mercaptoethanol, and 20 mM HEPES-NaOH (pH 7.8)), with the addition of one tablet of Pierce complete EDTA-free protease inhibitor and

phenylmethylsulfonyl fluoride (to 1 mM) to minimize proteolysis, and 40 mg/L of BV (Frontier Scientific) to ensure maximal chromophore occupancy. The lysates were clarified by sedimentation at $25,000 \times g$ for 25 min, and resulting supernatants were applied to a 30-mL column of Ni-NTA beads (Qiagen) equilibrated with Ni-NTA-binding buffer. Bound protein was eluted with binding buffer containing Pierce protease inhibitor cocktail and supplemented to a final concentration with 300 mM imidazole. *Ps*BphP1-containing fractions (~25 mL) were exchanged using a 200-mL G-25 Sephadex column into Ni-NTA-binding buffer to remove the effects of the protease inhibitor followed by incubation of the eluant for 3 h with 1 mg of TEV protease to remove the 6His-TEV tag. The released *Ps*BphP1 was then passed through a Ni-NTA column equilibrated with Ni-NTA-binding buffer to remove non-cleaved *Ps*BphP1, TEV protease, and high affinity contaminants. The samples were then exchanged as above by SEC into anion exchange binding buffer (10% (v/v) glycerol, 30 mM NaCl, 1 mM Na$_2$EDTA, and 30 mM HEPES-NaOH (pH 7.8)), and fractionated by FPLC with a 10 mL Q-Sepharose HP column (Cytiva) pre-equilibrated with the same buffer, and eluted with a linear 30–500 mM gradient of NaCl over 20 column volumes. Fractions containing non-tagged *Ps*BphP1 were concentrated with 30,000 MWCO centrifuge filters (Millipore), and flash frozen directly into liquid nitrogen as 30 μL droplets.

The coding sequence of full-length *Ps*AlgB was enriched by PCR from DNA extracted from *P. syringae* DC3000 cells and placed by Gibson assembly[41] into a modified pBAD vector in frame with an N-terminal 6His-TEV tag, and expressed under the control of the *araBp* promoter. The Asp59-Ala mutation was introduced into the resulting plasmid by Gibson assembly[41]. The *Dr*BphR coding sequence in frame with an N-terminal T7 tag and a C-terminal 6His tag was expressed under the control of the *lac* operon in a pET21a vector as described[28]. Both cultures were grown in terrific broth to an OD$_{600}$ of 1, and the incubation temperature lowered to 16 °C. After 2 h, protein expression was induced with 2 g/L arabinose for *Ps*AlgB and 1 mM IPTG for *Dr*BphR. As above, cultures were incubated for an additional 16 h at 16 °C, harvested by centrifugation, frozen directly in liquid nitrogen, and stored at −80 °C.

*Ps*AlgB was purified from frozen cells as described above for *Ps*BphP1 except that BV was not added during lysis and that Q-Sepharose HP anion exchange chromatography preceded Ni-NTA subtraction chromatography after TEV cleavage to enable appropriate buffer exchange prior to SEC. T7/6His-tagged *Dr*BphP was first enriched by Ni-NTA chromatography followed directly by Q-Sepharose HP chromatography and a final concentration with 10,000 MWCO centrifuge filters. Prior to freezing, the protein was exchanged into enhanced solubility buffer by SEC with a 23-mL Superdex 200 Increase 10/300 GL column.

To test for sample integrity, *Ps*BphP1 as Pr, *Ps*AlgB, and *Dr*BphR were exchanged into enhanced solubility buffer (175 mM KSCN, 10 mM 2-mercaptoethanol, and 10 mM HEPES-NaOH (pH 7.5)) using a Superdex 200 Increase 10/300 GL column (Cytiva). The samples were then assessed by SDS-PAGE followed by staining of the gels either for protein with Coomassie Blue, for bound BV in *Ps*BphP1 by zinc-induced fluorescence under UV light[29], and for autophosphorylation activity as described below.

## Cryo-EM grid preparation and data collection

Prior to cryo-EM data collection, *Ps*BphP1 was exchanged into enhanced solubility buffer as above, diluted to a concentration of 0.5 mg/mL, and either incubated in darkness for Pr or irradiated for 10 min with 630-nm LED at a fluence rate of 13.5 μmole m$^{-2}$ s$^{-1}$ to saturate photoconversion to Pfr. Cryo-EM grids were prepared using a FEI Vitrobot (Mark IV), with the chamber temperature set to 4 °C and the humidity set to 95%. Quantifoil R2/2 300 mesh copper grids were pretreated by either plasma cleaning (hydrophilic and negative charge) or glow discharging in the presence of amylamine

(hydrophobic and positive charge). The plasma-cleaned grids were treated for 1 min with a H$_2$/O$_2$ plasma mixture using the Gatan Solarus Plasma System (Gatan). The amylamine-treated grids were first negatively glow discharged for 60 sec in the presence of air with the copper side facing up using a GloQube Glow discharge system (Quorum Technologies). Grids were then flipped to carbon side up and positively glow discharged in the presence of amylamine (20 μL connected to inlet) for 60 s. Under dim green safelights, 3-μL aliquots of *Ps*BphP1 either as Pr or Pfr were pipetted onto the pretreated Quantifoil grids and then blotted for 2 s with blot force set to 0 (random units) after a 20 s wait time. The blotted grids were plunge frozen into liquid ethane, and stored in liquid nitrogen.

Cryo-EM images were recorded with a Falcon 4 direct electron detector (ThermoFisher Scientific) at a magnification of 96,000x corresponding to a pixel size of 0.657 Å at the specimen level. The datasets were collected automatically with EPU software using a Titan Krios microscope (ThermoFisher Scientific) operating at 300 kV. Stage tilts from 0 to 30° were used to enhance the angular distribution. Movies were recorded with a total dose of 51.78 or 50.54 electrons per Å$^2$ and an exposure time of 4.28 s. The objective lens defocus values were set to vary from −0.8 to −2.4 μm.

## Image processing

Acquired movies were motion corrected using patch motion correction in CryoSPARC v4.2.1[42]. CTF estimation was then performed on the subsequent micrographs[43]. Three data sets for Pr (plasma-cleaned, amylamine-treated, and plasma-cleaned with tilts) and three for Pfr (plasma-cleaned, amylamine-treated, and amylamine-treated with tilts) were independently blob picked and processed through 2D classification. Templates were generated from 2D classes and used for template picking and further 2D classification.

Particles chosen from 2D classes of the Pr data sets were processed through ab initio reconstruction and heterogenous refinement with multiple classes. See Supplementary Figs. 3–5 for the cryo-EM workflows. 380,203 Pr particles from the highest quality heterogeneous refinement volumes of the plasma-cleaned and amylamine-treated data sets were combined for non-uniform refinement using C2 symmetry. Particles from the plasma-cleaned with tilts collection were then merged for a total of 833,049 particles available for non-uniform refinement with C2 symmetry, which generated a 2.9-Å resolution Pr density map with reduced orientation anisotropy. To potentially resolve symmetry-breaking features, the particle stack was symmetry expanded around the C2 axis; subsequent local refinement of the resulting 1,666,098 total particles yielded a PSM reconstruction with a final gold standard resolution of 2.81 Å (PDB 8U4X, EMD-41903). The final Pr map was sharpened using DeepEMhancer[44] in the 'tight-Target' mode and was used for model building and refinement. To potentially resolve more of the DHp, 3DVA was performed with a low-pass filter of 5 Å with 3 sets orthogonal principle modes to solve[42]. One of the components represented variability in the length and direction of the extended DHp (Supplementary Fig. 3), and particles were clustered into 5 groups based on this specific component. The largest cluster with the best resolved extended DHp domain (119,594 particles) underwent local refinement, resulting in a 3.3-Å reconstruction (PDB 8U8Z, EMD-42030).

For the Pfr data sets, a combined total of 1,029,141 particles were chosen from 2D classification of the plasma-cleaned, amylamine-treated, and amylamine-treated with tilts data sets. Ab initio reconstruction was performed with 5 classes on all particles. Two classes were selected to be processed independently. One class consisting of 281,451 particles produced a 3.13-Å density map from local refinement after symmetry expansion around the C2 axis and was designated "medial" Pfr (PDB 8U65, EMD-41943). A separate class of 232,298 particles produced a 3.04-Å density map from local refinement after symmetry expansion around the C2 axis and was designated "splayed"

Pfr (PDB 8U66, EMD-41944). The remaining 537,192 particles from three classes underwent further heterogeneous refinement with an added 'junk' class to remove lower quality particles. Non-uniform refinement of 402,598 particles selected from the highest quality heterogeneous class generated a Pfr map that was designated DoD. Non-uniform refinement of the full-length DoDs produced a 3.3-Å density map (PDB 8U62, EMD-41941).

To better resolve the PSM as Pfr in the DoD datasets, signals from the HK bidomains were subtracted from the full-length DoD particle stack; the particles were then symmetry expanded around the C2 axis. Local refinement of the PSM DoD yielded a 3.01-Å density map (PDB 8U63, EMD-41942), which was sharpened using DeepEMhancer[44] in the 'tightTarget' mode. To generate the low-resolution reconstruction of the full tetrameric form (Supplementary Fig. 5), the DoD particle stack was first down-sampled to 1.314 Å/pixel and re-extracted to a larger box size. (400 pixel, or 525.6 Å). Duplicate particles were then removed using a minimum separation distance of 250 Å. After an initial non-uniform refinement with C2 symmetry, the particle stack was symmetry expanded followed by local refinement.

### Negative stain electron microscopy

*Ps*BphP1 samples were exchanged by SEC into enhanced solubility buffer or an NaCl buffer containing 175 mM NaCl, 10 mM 2-mercaptoethanol, and 10 mM HEPES-NaOH (pH 7.5), and diluted to a final concentration of 0.025 mg/mL. The samples were either incubated in darkness for Pr or irradiated to saturating levels of Pfr with a 630-nm LED as above. Under dim green safelights, 3-μL aliquots were absorbed for 60 s onto carbon-coated 200-mesh copper grids (01840-F, Ted Pella, Redding, CA), which had been glow discharged for 30 s in a Solarus 950 plasma cleaner (Gatan, Peasanton, CA). After incubation, each grid was washed five times with ultrapure water and stained with freshly prepared 0.75% (w/v) uranyl formate for 2 min. Excess uranyl formate was blotted off with filter paper (Whatman No.2, FisherScientific) before air drying.

Samples were imaged with a JEOL JEM-1400Plus transmission electron microscope (JEOL USA) at an operating voltage of 120 kV with a NanoSprint15-MkII 16-megapixel sCMOS camera (Advanced Microscopy Techniques) at 120 kV. Micrographs were collected as tiff files at a nominal magnification of 30,000x and a pixel size 3.54 Å/pixel. The micrographs were first converted to mrc files with inverted contrast using the e2proc2d.py script from EMAN2[45]. The rest of the data processing was done with CryoSPARC v4.2.1[42]. Briefly, particles were automatically picked, extracted with a 100 pixel box size, and subjected to 2D classifications into 100 classes. The best class averages were used as templates to re-pick the images, yielding higher-quality particles. After several rounds of 2D classification, the most informative classes were selected for comparison.

### Model building, refinement and validation

Model building was initially conducted for the Pr form of *Ps*BphP1. Amino acid residues 24−314, 334−443, and 478−501 from the X-ray crystallographic structure of the PSM from *Dr*BphP (PDB: 4Q0J)[29] were docked into the EM-density using rigid body refinement in Phenix[46] and served as guides for manual placement of *Ps*BphP1 residues. The model was refined iteratively via cycles of manual modeling in Coot[47] and real-space refinement with Phenix version 1.21rc1-4903[46]. Modeling of Pfr was facilitated by separate placement of the PAS-GAF and PHY (without the hairpin) domains from the Pr structure into the corresponding Pfr maps using Phenix rigid body refinement and an iterative strategy of manual modeling in Coot and refinement in Phenix. Refinement of the Pr model based on the 3.4-Å resolution map that focused on the DHp domains utilized the 2.8-Å resolution Pr model as a reference to restrain PSM geometry. Structural figures were prepared in UCSF Chimera[48], UCSF ChimeraX-1.2.5[49], and PyMol (https://pymol.org/2/).

### Sequence and structural alignments

Sequence alignments were conducted using Clustal Omega with default parameters[50]. The structure of the CA domains from *Ps*BphP1 were predicted from amino acid residues 594-745 using the RoseTTAFold modeling method[34] available from the Robetta protein structure prediction server (https://robetta.bakerlab.org). This model matched closely to CA or CA-like domains from over 100 proteins from the RCSB Protein Data Bank as determined by the DALI server (ekhidna2.biocenter.helsinki.fi/dali)[35]. Superpositions of PHY domains were conducted with PyMol (https://pymol.org/2/) using 3D models available in the RCSB Protein Data Bank. Portions of the superposed models correspond to the sequences found in Supplementary Fig. 12b or d.

### UV-vis absorption spectroscopy

UV-visible absorption spectroscopy was conducted in triplicate at 22 °C using a Cary60 spectrophotometer (Agilent). Prior to analysis, *Ps*BphP samples were exchanged into enhanced solubility or NaCl buffers using a Superdex 200 Increase 10/300 GL SEC column (Cytiva), and diluted to an $OD_{700 nm}$ between 0.3 and 0.4 as Pr (-0.4 mg mL$^{-1}$). Pr spectra were collected from dark-adapted samples whereas Pfr spectra were collected after a 10-min irradiation with a 630-nm LED at a fluence rate of 13.5 μmole m$^{-2}$ s$^{-1}$ to saturate photoconversion. For spectral analyses, the solution background and Pr and Pfr spectra were each collected in triplicate. Thermal reversion rates were measured in triplicate using absorption scans from 500 to 900 nm at a scan rate of 24,000 nm min$^{-1}$. Data were fit to a sum of two exponentials and nonlinear least squares fitting in R based on the equation: $Abs = Amp_1(exp(-k_1 t)) + Amp_2(exp(-k_2 t)) + Abs_0$, where Abs and $Abs_0$ were the measured absorbance and calculated baseline absorbance, respectively, $Amp_i$ and $k_i$ represent the $i$th amplitude and rate constant, and t is time. For each sample, $k_1$ and $k_2$ were fit globally and the parameters $Amp_1$, $Amp_2$, and $Abs_0$ were allowed to float. Generally, all wavelengths from 650 nm to 785 nm were included in the fit.

### Autophosphorylation and phosphotransfer

Five hundred-μL samples of *Ps*BphP1 or *Ps*AlgB were exchanged into kinase assay buffer (20 mM HEPES-NaOH (pH 7.5 at 22 °C), 5 mM $MgCl_2$, 0.2 mM $Na_2$EDTA, 10 mM 2-mercaptoethanol, and either 175 mM NaCl or 175 mM KSCN) by SEC using a Superdex 200 Increase 10/300 GL column or a 10-mL G-25 Sephadex column, respectively. Autophosphorylation assays were conducted by combining 1:1, a solution containing 2 μM *Ps*BphP1 and 1.5 μM BSA in kinase assay buffer with a solution of 300 μM ATP, 0.2 μCi μL$^{-1}$ γ-[$^{32}$P]-ATP and 1.5 μM BSA in kinase assay buffer, and assayed under saturating red light provided by a 660-nm LED (Thorlabs) (mostly Pfr). Reactions were quenched with hot SDS-PAGE sample buffer and subjected to SDS-PAGE. The gels were fixed in 40% methanol, dried, and quantified for autophosphorylation with an Amersham Typhoon phosphorimager combined with phosphor imaging plates (GE Health Care BioSciences). Signals were quantified using ImageQuant and data were fit to either a single exponential model, $S_t = ΔS \cdot (1-exp(-kt)) + S_0$, where $S_t$, $ΔS$, and $S_0$ are the counts at a given time, the amplitude and baseline values of the signal, respectively, k is the rate constant and t is time, or the sum of two such exponentials. The rate constant was fit globally, while all other parameters were allowed to float. Data analyses were conducted using the R software package (https://www.r-project.org).

Phosphotransfer to *Ps*AlgB was conducted in a similar manner (but without BSA) in kinase assay buffer containing 175 mM KSCN. *Ps*BphP1 at 8 μM was combined in a 1:1 volumetric ratio with assay buffer containing 300 μM ATP spiked with γ-[$^{32}$P]-ATP (0.2 μCi μL$^{-1}$). This mixture was incubated for 2 h at 22 °C under saturating red light to allow maximal loading of the HK phosphoacceptor. The solution was then combined in a 1:1 volumetric ratio with assay buffer containing 2 μM *Ps*AlgB. Aliquots were quenched, separated by SDS-PAGE, and analyzed for $^{32}$P as above. *Ps*BphP1 harboring a His530-Ala

substitution of its acceptor histidine and *Ps*AlgB harboring the Asp59-Ala substitution of its receiver aspartate were assayed as above. Data fitting was conducted in the same manner as the autophosphorylation assays, except that baselines were constrained to the values measured at time zero.

### Size exclusion chromatography (SEC)

SEC at 22 °C used *Ps*BphP1 samples exchanged into enhanced solubility buffer and either kept in darkness as Pr or photoconverted to Pfr with saturating red light. One hundred-μL samples at 3 mg mL$^{-1}$ were immediately loaded onto a Superdex 200 Increase 10/300 GL column coupled to an AKTA FPLC (GE) (flow rate of 0.5 mL min$^{-1}$), and subjected to SEC either in darkness (Pr) or under continuous irradiation with a circular array of 72 630-nm LEDs (superbrightleds.com, part number STN-BRED-A6A-10B1M-24V) to maintain saturating levels of Pfr. Elution was monitored at OD$_{280}$ using the Unicorn software (Cytiva) and verified by SDS-PAGE of collected fractions.

The absorption contributions of *Ps*BphP1 dimers and tetramers were estimated by fitting the data to the sum of two exponentially modified normal distributions, using the equation, Abs = 0.5 Amp$_T\lambda_T$exp(0.5 $\lambda_T(2\mu_T + \lambda_T\sigma_T^2 - 2V_e)$)erfc(($\mu_T + \lambda_T\sigma_T^2 - V_e$) $(2^{0.5}\sigma_T)^{-1}$) + 0.5 Amp$_D\lambda_D$exp(0.5 $\lambda_D(2\mu_D + \lambda_D\sigma_D^2 - 2V_e)$)erfc(($\mu_D + \lambda_D\sigma_D^2 - V_e$)$(2^{0.5}\sigma_D)^{-1}$) + b + cV$_e$. Here, Amp serves as a proportionality constant for signal magnitude of dimers (D) or tetramers (T), V$_e$ is elution volume, b and c are parameters to model linear drift of the absorption baseline, erfc is the complementary error function, σ is the square root of the variance, and λ and μ are parameters for modeling the skew of the data. While these measurements did not represent the relative abundances of a system at equilibrium, we used an equilibrium equation to show a trendline as a function of initial *Ps*BphP1 concentration: $\bar{Y} = 2(4K_2C_{tot} + 1 - (1 + 8K_2C_{tot})^{0.5})(8K_2C_{tot})^{-1}$, where $\bar{Y}$ is the mole fraction of tetramers, K$_2$ is the assembly equilibrium constant, and C$_{tot}$ is the total concentration of subunits in M units.

SEC assays for the ATP-dependent binding of *Ps*BphP1 to *Ps*AlgB or *Dr*BphR used the enhanced solubility buffer and chromatographic conditions as above with the addition of 5 mM MgCl$_2$ and 0.2 mM Na$_2$EDTA, with or without 0.15 mM ATP. To bias formation of DoDs, *Ps*BphP1 samples were added to a final concentration of 3 mg mL$^{-1}$; the response regulators were used at 3 mg mL$^{-1}$. To maintain sample stability at high concentrations, *Dr*BphR was stored at 16.1 mg mL$^{-1}$ in enhanced solubility buffer, while *Ps*AlgB and *Ps*BphP1 were stored in the Ni-NTA binding buffer at 20 mg mL$^{-1}$ and in anion exchange buffer at 8 mg mL$^{-1}$, respectively. Samples were combined and incubated for 10 min either in darkness or while being irradiated to saturation with a 630-nm LED. For samples examined in running buffer containing ATP, ATP was added directly to the sample to a concentration of 1.5 mM and incubated for 10 min in darkness or under red light as indicated. As above, the Pfr samples were continuously irradiated with 630-nm LEDs during chromatography. Microsoft Excel was used for all SEC analyses requiring data fitting to a model.

### Reporting summary

Further information on research design is available in the Nature Portfolio Reporting Summary linked to this article.

## Data availability

The 3D cryo-EM map of full-length *Ps*BphP1 as Pr has been deposited in the Electron Microscopy Data Bank database under accession codes EMD-41903. The corresponding atomic model has been deposited in the RCSB Protein Data Bank (http://www.rcsb.org) under accession code PDB-8U4X [https://www.rcsb.org/structure/8U4X]. The lower resolution map and model of Pr with an extended view of the DHp domain are available in the Electron Microscopy (https://www.ebi.ac.uk/emdb) and RCSB Protein Data Banks under accession codes EMD-42030 and PDB-8U8Z [https://www.rcsb.org/structure/8U8Z],

respectively. The 3D cryo-EM maps of *Ps*BphP1 as Pfr used to generate the full-length DoD, the DoD PSM, the medial PSM, and the splayed PSM models are available in the Electron Microscopy Data Bank database under accession codes EMD-41941, EMD-41942, EMD-41943, and EMD41944 [https://www.ebi.ac.uk/emdb/EMD-41943], respectively. The corresponding atomic models for the four Pfr maps have been deposited in the RCSB Protein Data Bank under accession codes PDB-8U62 [https://www.rcsb.org/structure/8U62], PDB-8U63 [https://www.rcsb.org/structure/8U63], PDB-8U64 [https://www.rcsb.org/structure/8U64], and PDB-8U65 [https://www.rcsb.org/structure/8U65], respectively. The 3D structure of the PSM from *Dr*BphP[29] was obtained from the RCSB Protein Data Bank under the accession code PDB 4Q0J. Source data are provided with this paper.

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

## Acknowledgements

Cryo-EM data were collected with a ThermoFisher Scientific Titan Krios microscope at Washington University in St. Louis Center for Cellular Imaging. This work was funded by the US National Institutes of Health R01 grant GM127892-05 (to R.D.V.) and by funds provided by Washington University in St. Louis (to R.D.V. and K.B.).

## Author contributions

E.S.B., K.B., M.J.R., J.A.J.F. and R.D.V. designed the experiments. E.S.B., A.J.M. and V.G. expressed and purified the *Ps*BphP1, *Ps*AlgB, and *Dr*BphP samples. E.S.B. performed the spectroscopic, SEC, and phosphotransfer assays. K.B., M.J.R. and B.S. performed the cryo-EM and 3D reconstructions of *Ps*BphP1. E.S.B. built and refined the atomic models. K.B. conducted the negative-staining cryo-EM. E.S.B., K.B. and R.D.V. wrote the manuscript with input from all other authors.

## Competing interests

The authors declare no competing interests.
