## [Peer Review File · Nature Communications]

Signaling by a Bacterial Phytochrome Histidine Kinase Involves a Conformational Cascade Reorganizing the Dimeric PhotoreceptorREVIEWER COMMENTS

Reviewer #1 (Remarks to the Author):

Burgie et al. here use cryo-EM and a variety of biophysical, biochemical, and computational models to advance our understanding of the light-induced conformational changes associated with the Pr/Pfr change in the bacterial histidine kinase PsBphP1 from *Pseudomonas syringae*. Using bacterially-expressed protein incorporating biliverdin *in vivo*, the author first investigate the structural state of Pr, finding a dimeric assembly highly reminiscent of prior crystal structures of PAS-GAF-PHY photosensory modules. While some segments of the DHP helices connecting the PSMs to the HKs were observed, the HKs themselves were not seen and required modeling to suggest likely positions. The Pfr structural information chiefly focused on a tetrameric “dimer of dimers” (DoD) model with one of the two catalytic CA domains in an unproductive conformation; a second CA domain was modeled to be elsewhere. Conformational changes between two states are reminiscent of those proposed on simpler systems (i.e. photosensory modules) by this group and others. Lastly, the authors use phosphotransfer assays to confirm PsAlgB as the cognate response regulator in this system and provide evidence for interaction through a transient complex.

Overall, this paper reasonably accomplishes its aim of furthering understanding of phytochrome signaling mechanisms, though clarification is needed on a few points:

Major points:

1. Neither the Summary nor Introduction sections specify that models were generated using a combination of Cryo-EM data and structural predictions. I strongly suggest that a disclaimer similar to that seen on lines 405-407 should somehow be placed in these sections so readers are crystal clear as to what is established from experimental data and what isn't (i.e. CA domain locations in several of the structural models).
2. In the Discussion section, the authors distinguish this as “one of the first near complete 3D models of the native Pr and Pfr conformers...” (line 401), but do not cite the implied comparable ones. These should be explicitly cited here. On a related note, discussion on line 414 mentions that certain steps of the proposed mechanism ascertained by these models “have been seen before” (line 414), but these too are somewhat vaguely identified. What do that authors view as their unique contribution?

Minor points:

- throughout: I encourage the authors give a thorough readthrough to ensure abbreviations are defined and standard in the field, e.g. line 64: nPAS vs. PAS; Per/Arndt/Sim -> Per/ARNT/Sim; PSM used w/o definition line 68 ...
- line 274 Unclear how the second CA domain was placed here— is there any density there to suggest its presence?
- p. 20 “Sequence and structural alignments” section— Were any other CA domains closely related to that of PsBphP1 aligned with the predicted RoseTTAFold model?
- p. 21 “Autophosphorylation and phosphotransfer” section— What was the final concentration of PsBphP1 and PsAlgB in the autophosphorylation and phosphotransfer assays?

Reviewer #2 (Remarks to the Author):

Based on EM data of the full length bacterial phytochrome, the authors provide details of a Pr Pfr modulation of histidine kinase activity. These results extend our knowledge of intramolecular signal transduction of the phytochrome photoreceptors, it adds up to the many valuable contributions of the authors on crystal and EM structures of phtychromes or different kind. It is interesting that EM measurements were made with PsBphP, whereas DrBphP that was discovered and used for crystal structure of phytochrome fragments was now analyzed by EM by another group. Apparently, histidine

kinases are flexible, in contrast to the PSM. Therefore only part of HK is resolved. The authors used modeling programs to fill up the parts that are not resolved. These parts should be better discussed regarding reliability. If only for visualization then this should be stated more clearly. If conclusions drawn from alphafold or Rosetta, this should be mentioned where the conclusions are discussed.

Specific points are noted below, with the relevant line number:

45 why speak of kingdoms, and if you do, you should say "all kingdoms". It is the organism that contains photoreceptors, not kingdoms

64 why nPAS, why the n? Makes sense for plant phytochromes, but not if there is only one PAS

135 what buffers

149 > 80%, what does it mean, between 80 and 100? Does it vary? Is it not exactly known? How was it determined? Further down where the EM data are explained, where are the 20% Pr remaining?

Pfr/Pfr homodimers should have 64%, and the heterodimers 16%. Are these Pr and heterodimers somehow sorted out in the analysis or overruled? Should be discussed in the context of EM with Pfr.

170 what moves as a pair and when is DIP indeed invariant?

181 as I understand, the missing hydrogen bond is in Pr, why should this explain the instability of Pfr? Dark reversion is certainly affected by many interactions, and this is probably not the only difference between Dr and Ps

204 ff I find it critical to mix EM and computer models. The experiments were made to get information superior to the models, otherwise it had not to be made. Therefore, the alphafold and Rosetta modeling should be better explained and discussed: how do the models fit with each other and with the EM structure? How were the transitions handled? One would expect that the results are dependent on the alignment between EM and model. What exactly was the reason to use the models? Was it just for visualisation or are the interpretations also dependent on the use of the models? How trustable is the proposed mechanism under inclusion of computer models? Alphafold produces several predictions, only one is in the database. Did you try to get the others? It also had to be said that there was probably no chromophore in the models and that there is no Pr/Pfr.

If the models were only used for presentation, this should be made clearer.

344 Fig. 3Ab should show DoD, but it does not (?)

254 The = the

255 also the hairpin could indicate real Pfr, but this comes later

261 I did not understand the sentence starting with compensating

265 same for sentence starting with GAF

277 sentence combines too many information and is unclear; both statements should have an explanation

283 the term phosphotransfer for ATP to His or from His to Asp?

299 the first is ... and the second? Better start new sentence before „the“

303 what are the four? From either subunit of either dimer?

313-317 in my view, these theories are too speculative. "buffer large motions of PHY": in Agp1 the PSM forms weak dimers which can dissociate into monomers, meaning that the full length dimer is held together by the histidine kinase. A pivot between GAF and PHY would not stabilize subunit interaction of GAF.

"whose function remains enigmatic" is just a rhetoric statement, as if now the function would be clear, how should a backstop to pivoting look like?

326-330 what is the evidence for this assumption? Note that many bacterial phytochrome histidine kinase are down regulated upon Pr to Pfr transition. This should be mentioned in the introduction and considered in the discussion when it comes to universal statements.

332-337 DoD .. telling about the mechanism, sorry, I did not understand what is meant

337 why is it a key aspect that contact is transient?

351 avoid such challenges, was this really the motivation? Without fusion would rather be the standard approach.

362 and following, if I understand it right, there is no specificity for response regulator. On the other

hand, it is often spoken of cognate response regulators, kinase and regulator are in the same operon. What about Pr-Pfr in trans phosphorylation (see below)?

365 was the His-tag at the N-terminus? Fig. S1, where is the Gly remnant?

367 EM was made for Pr Pfr, were there any Pr Pfr comparisons with respect to autophosphorylation or phosphotransfer?

373, to accept P is certainly not dependent on the strength of labeling

379 SEC can be used to detect protein interactions, if they are strong. How strong does an interaction have to be for SEC to be detected? In other words, if there is no effect in SEC, it does not mean that there is no interaction. For Agp1/Agp2 interaction we found: no effect on SEC, effect on phosphorylation, effect on spectra, effect on FRET. The statement „failed to bind“ is not correct, because it is not known that it does not bind. Maybe „failed to affect mobility of BphP on SEC columns“. Please check also abstract. See also 384

394 interesting finding on the stimulation of phosphorylation , it would be good if it could be investigated further.

416 ZZZssa to ZZEssa. Here comes a mixture of known results and the present data. This is ok for the discussion but should also be named as such (line 413 ..) . I would prefer if would be clear what was already known or suggested and what is new. For example that first step is the isomeriation it clear, but is not the result of the present work.

418 selection between two rotational orientations of the PHY domain. I do not have in mind whether this is described in the results section, but it could be explained here again.

419 prying force .. translates .. helical spine : where is the prying force, what kind of motions. Which feature spans via ... , the prying force, the translation', the movement of helices? Could be more precise

430 why is it an advantage of EM, there are not necessarily different forms.

441: adjustment, are they not adjusted? Is there evidence that dimer contacts need to be adjusted? Why complete dissociation? For me, the sentence was distracting.

460, one would like to hear some words here or elsewhere that many bacterial phytochrome histidine kinases have a strong autophosphorylation in the Pr and weaker in the Pfr.

461 helps explains → helps explain

466 a second what, first is missing

505 why does decrease in temperature result in enhancement of heme synthesis, even before HO is induced?

507 also here, lowering of temperature for induction of expression?

510 funny, but does your centrifuge have green light inside? Centrifugation force and time or refer to a previous work

514 fourfold of what

516 518 was PMSF added twice?

524 diameter and length of Sephadex is important, also the loading volume. For good separation, loading volume should be 1% of column volume, this would be 2 ml. For buffer exchange, more is possible but in any case, one would like to know the volume

530 good luck with the Q Sepharose, in our case it denatured the phytochromes

532 and 533, what was the protein concentration

up to 556, would be good if protein concentrations would be given more often

560 and 571, different concentrations given for the same sample

479 detectable

483 remove „a“

622 DoD PSM

624 check for double blanks

680 to get a general impression of HK folding: were also other HK structures compared? How many are there in the database? How similar is the second hit ? Etc. In general, the folding should be better explained in the results section: what parts were folded,

690, loading volume
706 column and sample volumes? As above or different? Which was used when?
711 fixing solution should contain acetic acid, in methanol only, the proteins can diffuse, but ok, apparently it worked
718 for what was Excel used and for what R
731 volume of sample and column dimensions important
736 what is Unicorn

Here are some contributions from our group that you can consider if you like

[1] helical pine
[2] [3] PSM dimer and monomer, flexibility of HK in proteolysis
[4] phytochrome interaction, SEC
[5] no distance change from Pr to Pfr
[6] no flexibility shift from Pr to Pfr

1. Nagano, S., Scheerer, P., Zubow, K., Michael, N., Inomata, K., Lamparter, T. & Krauss, N. (2016) The crystal structures of the N-terminal photosensory core module of *Agrobacterium* phytochrome Agp1 as parallel and anti-parallel dimers, *J Biol Chem.* 291, 20674-20691.
2. Noack, S., Michael, N., Rosen, R. & Lamparter, T. (2007) Protein conformational changes of *Agrobacterium* phytochrome Agp1 during chromophore assembly and photoconversion, *Biochemistry.* 46, 4164-76.
3. Esteban, B., Carrascal, M., Abian, J. & Lamparter, T. (2005) Light-induced conformational changes of cyanobacterial phytochrome Cph1 probed by limited proteolysis and autophosphorylation, *Biochemistry.* 44, 450-461.
4. Xue, P., El Kurdi, A., Kohler, A., Ma, H., Kaeser, G., Ali, A., Fischer, R., Krauss, N. & Lamparter, T. (2019) Evidence for weak interaction between phytochromes Agp1 and Agp2 from *Agrobacterium fabrum*, *FEBS Lett.* 593, 926-941.
5. Kacprzak, S., Njimonu, I., Renz, A., Feng, J., Reijerse, E., Lubitz, W., Krauss, N., Scheerer, P., Nagano, S., Lamparter, T. & Weber, S. (2017) Intersubunit distances in full-length, dimeric, bacterial phytochrome Agp1, as measured by pulsed electron-electron double resonance (PELDOR) between different spin label positions, remain unchanged upon photoconversion, *J Biol Chem.* 292, 7598-7606.
6. Elkurdi, A., Guigas, G., Hourani-Alsharafat, L., Scheerer, P., Nienhaus, G. U., Krauss, N. & Lamparter, T. (2023) Time-resolved fluorescence anisotropy with Atto 488-labeled phytochrome Agp1 from *Agrobacterium fabrum*, *Photochem Photobiol.*

Reviewer #3 (Remarks to the Author):

Phytochromes are a superfamily of red/far-red photosensing proteins in diverse species and regulate downstream signalling pathways. The manuscript by Burgie et al. aims to elucidate the mechanistic basis of photoconversion by solving the resting (Pr) and the photoactivated (Pfr) structures of a bacteriophytochrome. To this aim, the full-length, biliverdin-bound bacteriophytochrome BphP1 from *P. syringae* (PsBphP1) was expressed and purified. Structures at overall resolutions between 2.8 Å and 3.4 Å of the Pr and Pfr states were obtained. The authors provide based on these structures, a mechanism how the configuration of the biliverdin allosterically translates to the interconnected histidine kinase (HK) domain, thereby increasing its autokinase activity and affecting the cognate response regulator AlgB. For the Pfr state, medial, splayed and tetrameric (dimer-of-dimer) configurations were obtained. For the Pr, a single conformation was obtained. Unresolved/not interpretable features within obtained Coulomb potential maps, e.g., the catalytic ATPase domains

(CA), were modelled by a predicted AlphaFold model, providing a hypothetical structure of the dimeric full-length protein.

Numerous structures of BphPs (although some are only fragments) were published previously and an allosteric communication mechanism was reported (PMID: 36509762) raising serious questions about the novelty of the here presented work. Furthermore and (also) importantly, I have serious concerns related to the resolution of the maps and the subsequent interpretation. While the photosensory modules (PSMs) are well resolved, as in previously published studies, the placement and structures of the effector domains rely on predicted, hypothetical models and fitting into non-existing (Fig. 4) and very low resolution density maps. This further weakens the claims and the novelty considerably.

Major points from above in more detail

- Since this manuscript does not provide the very first complete structures of a phytochrome in both Pr and Pfr states (the first reported is the phytochrome from *Deinococcus radiodurans* in 2022, PMID: 36509762), its novelty is limited. The structural consequences of the photoconversion in the PSM part is similar as reported previously.

- While the EM density of PSM and minor part of the S-helices are reasonable for the built model and made interpretation, care must be taken with overinterpretation of the weak/absent density and following modelling with AlphaFold models. This yields hypothetical 3D models of the different states. Therefore the made claims are ambiguous and problematic, e.g., Fig. 4.

- The finding that PsBphP1 as Pfr forms tetramers is remarkable and its dynamic equilibrium was subject of investigation. Unfortunately, it's very unclear/questionable from a technical standpoint how this map was obtained. Regarding the workflow in Suppl. Fig. 4: It seems as if the tetramer map was generated from symmetry-expanded dimer map with no ab-initio model generated before, which is problematic. As a minor issue, the local resolution for the tetrameric map as well as the orientation distribution plots for all final maps are not indicated.

- While previous structures are referenced, there is no comparison (except the BV overlay in Fig. 1b) between them and the structures solved here.

- Supplemental Fig. 3, panel c is misleading. The attributed resolution of 3.3 Å is difficult to understand considering the red density parts that would be expected to be less resolved than 4.5 Å. It would be important to show the mask used to obtain the indicated 3.3 Å.

Less major and minor point

- What is the rationale behind using the chaotropic reagent thiocyanate as a stabilizer in the "enhanced solubility buffer"? I agree that on a global scale the structure seems not to be affected, but locally it could disrupt interactions and contribute to higher mobilities of certain domains.

- Suppl. Fig. 2: To pretend that KSCN has no effect on the 3D structure of the PsBphP1 protein based on negative stain data (low resolution, heavy metal salt and specimen drying) and 2D class averages is not credible.

- The Reviewer suggest careful reading of the manuscript, here a few issues:

- line 518: PMSF is mentioned twice

- line 239: A bracket is missing

- line 281: punctuation mark

- Inconsistency: e.g., cryo-EM in line 231, cryoEM in line 430

- line 150: 2D class averages cannot be collected – please rephrase.

- Fig. S1: remove the text "Lorem ipsum" ...

- "Supplemental" should be replaced by "Supplementary"

- The authors might want to revisit model building: For example, in Supplemental Fig. 6, panel b, residues of S-helix(B) don't seem to fit well within the density.

Response to Reviewers' Comments:

The Reviewer comments are in *italic*. Our responses are in plain type and when appropriate quotations from the paper are in bold.

Reviewer #1 Comments

Burgie et al. here use cryo-EM and a variety of biophysical, biochemical, and computational models to advance our understanding of the light-induced conformational changes associated with the Pr/Pfr change in the bacterial histidine kinase PsBphP1 from Pseudomonas syringae. Using bacterially-expressed protein incorporating biliverdin in vivo, the author first investigate the structural state of Pr, finding a dimeric assembly highly reminiscent of prior crystal structures of PAS-GAF-PHY photosensory modules. While some segments of the DHp helices connecting the PSMs to the HKs were observed, the HKs themselves were not seen and required modeling to suggest likely positions. The Pfr structural information chiefly focused on a tetrameric “dimer of dimers” (DoD) model with one of the two catalytic CA domains in an unproductive conformation; a second CA domain was modeled to be elsewhere. Conformational changes between two states are reminiscent of those proposed on simpler systems (i.e. photosensory modules) by this group and others. Lastly, the authors use phosphotransfer assays to confirm PsAlgB as the cognate response regulator in this system and provide evidence for interaction through a transient complex.

Overall, this paper reasonably accomplishes its aim of furthering understanding of phytochrome signaling mechanisms, though clarification is needed on a few points:

Major points:

1. Neither the Summary nor Introduction sections specify that models were generated using a combination of Cryo-EM data and structural predictions. I strongly suggest that a disclaimer

similar to that seen on lines 405-407 should somehow be placed in these sections so readers are crystal clear as to what is established from experimental data and what isn't (i.e. CA domain locations in several of the structural models).

As requested, we further described that the *PsBphP1* structures were mainly generated by high-resolution cryo-EM and then augmented by lower resolution cryo-EM views and RoseTTAFold and AlphaFold structural predictions to generate a near full atomic view of the dimeric photoreceptor as Pr and Pfr. As stated above, this combined approach is now outlined in Abstract (lines 36-40), clarified at the end of the Introduction (lines 110-121), and described in detail in the Results (lines 216-220; lines 229-234, and lines 300-308).

2. In the Discussion section, the authors distinguish this as “one of the first near complete 3D models of the native Pr and Pfr conformers...” (line 401), but do not cite the implied comparable ones. These should be explicitly cited here. On a related note, discussion on line 414 mentions that certain steps of the proposed mechanism ascertained by these models “have been seen before” (line 414), but these too are somewhat vaguely identified. What do that authors view as their unique contribution?

As requested, we now more explicitly describe and reference at the beginning of the Discussion (Lines 437-444) the prior paired structures of full-length dimeric Phys, that being *Idiomarina* PadC and *X. campestris* BphP that terminate in a diquanylyl cyclase motif or a protein-protein interaction PAS9 motif of unknown function, respectively. We further place our structure in context of these prior models on lines 489-492. To our knowledge, the structures of native *P. syringae* BphP1 are the first to describe the paired models of a more common microbial Phy acting as a transmitter kinase without any modification or fusion. We also acknowledge and discuss on lines 500-510 the study of Wahlgren et al., (2022 *Nat. Commun.*) that presented paired models of *Deinococcus radiodurans* BphP in which the Phy polypeptide was translationally fused via a short linker to the C-terminus of its possible cognate phosphatase *DrBphR*. Our native model of *PsBphP1* differs from this fusion model, leading us to question whether the fusion accurately reflects the native situation. See comments in response to Reviewer #3 for more details.

Minor points by Reviewer #1:

Throughout: I encourage the authors give a thorough readthrough to ensure abbreviations are defined and standard in the field, e.g. line 64: nPAS vs. PAS; Per/Arndt/Sim -> Per/ARNT/Sim; PSM used w/o definition line 68 ...

Fixed the abbreviations as requested.

line 274 Unclear how the second CA domain was placed here— is there any density there to suggest its presence?

If we dropped the cryo-EM Pfr maps below 7.3-Å resolution, we could see faint signs of the second CA domain, but its positioning in the model was obscured by the other OPM seen within the tetrameric DoD map. Consequently, we just placed the domain model predicted by RoseTTAFold next to the DHp in roughly the same orientation as that of its protomer (which we could place) assuming the length of the connection to the DHp domain. We explained this manipulation in detail in the legend of Figure 4.

*p. 20 “Sequence and structural alignments” section— Were any other CA domains closely related to that of *PsBphP1* aligned with the predicted RoseTTAFold model?*

A number of CA and CA-like domains closely aligned with that of *PsBphP1* predicted by RoseTTAFold as determined by the DALI server. The CA domain from *Thermotoga maritima* HK 853(PDB ID 4JAV) determined by X-ray crystallography was one of the best matches. As it is a well characterized and well-cited structure, we chose it for comparison.

- p. 21 “Autophosphorylation and phosphotransfer” section— *What was the final concentration of PsBphP1 and PsAlgB in the autophosphorylation and phosphotransfer assays?*

As stated in the Figure Legends, both were used at 3 mg mL⁻¹

Reviewer #2 Comments:

Based on EM data of the full length bacterial phytochrome, the authors provide details of a Pr Pfr modulation of histidine kinase activity. These results extend our knowledge of intramolecular signal transduction of the phytochrome photoreceptors, it adds up to the many valuable contributions of the authors on crystal and EM structures of phytochromes or different kind. It is interesting that EM measurements were made with PsBphP, whereas DrBphP that was discovered and used for crystal structure of phytochrome fragments was now analyzed by EM by another group.

For historical perspective, our lab was the first to describe *DrBphP* and published an array of ground-breaking studies on this photoreceptor since 1999. But we felt it reached the end of its use given its poor kinase activity (in fact it is more likely a phosphatase from studies of the Multamaki et al. (2021) Nat. Commun. *P. syringae* BphP by contrast acts as a robust kinase. Now that we know in this report the identity of its cognate response regulator for *PsBphP1* with a defined output effect on quorum sensing, it provides a much better experimental model.

Apparently, histidine kinases are flexible, in contrast to the PSM. Therefore only part of HK domain is resolved. The authors used modeling programs to fill up the parts that are not resolved. These parts should be better discussed regarding reliability. If only for visualization then this should be stated more clearly. If conclusions drawn from alphafold or Rosetta, this should be mentioned where the conclusions are discussed.

As also requested by Reviewer #1, we described how the models were generated more explicitly in the Abstract (lines 36-40), clarified at the end of the Introduction (lines 110-121), and described in detail in the Results (lines 216-220; lines 229-234, and lines 300-308).

Minor Points by Reviewer #2

We have addressed all these minor points in the revised manuscript.

- 45 *why speak of kingdoms, and if you do, you should say “all kingdoms”. It is the organism that contains photoreceptors, not kingdoms.*

Fixed, to read “All”

- 64 *why nPAS, why the n? Makes sense for plant phytochromes, but not if there is only one PAS.*

Fixed, removed n.

- 135 *what buffers.*

Added Buffer (HEPES-NaOH).

- 149 > 80%, what does it mean, between 80 and 100? Does it vary? Is it not exactly known? How was it determined? Further down where the EM data are explained, where are the 20% Pr remaining? Pfr/Pfr homodimers should have 64%, and the heterodimers 16%. Are these Pr and heterodimers somehow sorted out in the analysis or overruled? Should be discussed in the context of EM with Pfr.

This 80% is just an estimate based on percent Pfr/Ptotal values determined for other phytochromes. Added “~80% of total Phy pool” for clarification.

Reviewer #2 is right that in the mix of PsBphP1 dimers there should be a significant percentage of Pr:Pfr heterodimers which were not evident as separate populations of particles. Presumably this low population was insufficient to produce a separate high quality map but if we collected ten times more data maybe they would become apparent. The issue of missing all the conformations of a given protein is just a fact for the current state of cryo-EM, which can hopefully be resolved at some point to help us better understand protein dynamics and allow statistical mechanics-level understandings of protein function.

- 170 what moves as a pair and when is DIP indeed invariant?

Removed the word “invariant” to avoid confusion.

- 181 as I understand, the missing hydrogen bond is in Pr, why should this explain the instability of Pfr? Dark reversion is certainly affected by many interactions, and this is probably not the only difference between Dr and Ps.

True, there is a missing hydrogen bond in Pr, but the (missing) histidine is an important anchor for the C-ring propionate in Pfr. This is compensated for PsBphP1 in part by the serine at position 284. See lines 281-287 for a discussion of this point. For H>A mutations in *Arabidopsis* PhyB, the formation of Pfr by with red light is blocked, while for *Deinococcus radiodurans* BphP photoconversion is significantly crippled. To clarify the importance “to hydrogen bond with” was changed to “to provide an important anchor for” and references were provided for the Wagner et al mutational study of *Deinococcus* BphP and the Burgie et al mutational study of PhyB.

- 204 ff I find it critical to mix EM and computer models. The experiments were made to get information superior to the models, otherwise it had not to be made. Therefore, the alphafold and Rosetta modeling should be better explained and discussed: how do the models fit with each other and with the EM structure? How were the transitions handled? One would expect that the results are dependent on the alignment between EM and model. What exactly was the reason to use the models? Was it just for visualisation or are the interpretations also dependent on the use of the models? How trustable is the proposed mechanism under inclusion of computer models? AlphaFold produces several predictions, only one is in the database. Did you try to get the others? It also had to be said that there was probably no chromophore in the models and that there is no Pr/Pfr.

If the models were only used for presentation, this should be made clearer.

As also requested by Reviewer #1 above, we described how the models were generated more explicitly in the Abstract, Introduction, Results, and Discussion. As examples see lines 36-40 in the Abstract, clarifications at the end of the Introduction in lines 110-121, and detailed descriptions in the Results on lines 216-220; lines 229-234, and lines 300-308.

For a more detailed explanation that was too long to be included in the text, it was indeed our goal was to account for every atom, but the native molecule in solution is very dynamic, which is very likely needed for its function. The data show clear residue-level data for almost all of the PSM. Still there are missing parts within the PSM due to

motion which obfuscate certain features, but we built what we can see. Once we get to roughly the phosphoreceptor histidines in the DHp domain of both the Pr and Pfr, the resolution drops dramatically. For Pr and more so with the Pfr DoDs the general outlines of the HK domains are visible but the resolution was insufficient for manual modeling.

Remarkably, secondary structure features were visible for the entire DHp region of all four DHp domains of the DoDs and two of the four CA domains. Here, the CA domain generated by RoseTTAFold provided an excellent match for the density (see the new Supplementary Fig 10). The primary AlphaFold model was used to estimate how the two helices of each DHp were juxtaposed. This was used as an initial guide for interpreting the density and the models of the DHps were manipulated by hand to fit the data. So, the AlphaFold model was essentially a starting point to direct manual model building of the portions of the DHp where side chains were not well defined by the density. For the Pr molecule, the CA domains show an outline of the tertiary structure consistent with the shape of the RoseTTAFold models. These were placed so the helices of the DHp and CA domains abutted each other as seen in every structure of HK bidomains to date. Positioning of the CA domain took into account the position of the DHp/CA domain linker which is distinctly visible in the Pfr dimer of dimer structure. Given that information other positions were untenable

- 244 *Fig. 3Ab should show DoD, but it does not (?)*

As far as we can see, Figure 3A,B does show cryo-EM views of the DoDs.

- 254 *The = the*

Fixed

- 255 *also the hairpin could indicate real Pfr, but this comes later*

We reworded this sentence on lines 272-275 to read: **“Importantly, the PSM regions of all three maps (DoD, medial, and splayed) unambiguously modeled both protomers as Pfr as judged by sufficiently resolved ZZEssa conformations of the bilins that aligned well for all three maps except for slightly different tilts of the D rings (Supplementary Fig. 9c) and by the presence of obvious α -helical hairpins, which are both signatures of this spectral state (Fig. 3e,f; Supplementary Figs. 7a and 9d-g).**

- 261 *I did not understand the sentence starting with compensating.*

We reworded the sentence on lines 281-284 for clarification of this point

- 265 *same for sentence starting with GAF.*

Add clarification to lines 287-290 to read: **“Another notable change in the GAF domain connections with the B-ring propionate was the use of Arg218 rather than Arg250 to anchor the propionate carboxylate group through a salt bridge”**

- 277 *sentence combines too many information and is unclear; both statements should have an explanation.*

As suggested we reorganized the paragraph on lines 298-313 to better explain the model building and subsequent ramifications of phosphotransfer.

- 283 *the term phosphotransfer for ATP to His or from His to Asp?*

Hypothesis now clarified on lines 311-313 to read: **“Either scenario would increase the local concentrations of histidine phosphoacceptors in the DHp domains and the catalytic ATP-binding sites in the CA domains to presumably accelerate ATP to histidine phosphotransfer.”**

- 299 the first is ... and the second? Better start new sentence before „the“

As recommended, we started a new paragraph on line 344-350 to explain the second possible consequence of the hairpin reconfiguration.

- 303 what are the four? From either subunit of either dimer?

Revised Lines 333-335 to read: **“or only the four PSM structures described here with PsBphP1 (i.e., one Pr and three Pfr models) (Supplementary Fig. 12c,d),”**

- 313-317 in my view, these theories are too speculative. “buffer large motions of PHY”: in Agp1 the PSM forms weak dimers which can dissociate into monomers, meaning that the full length dimer is held together by the histidine kinase. A pivot between GAF and PHY would not stabilize subunit interaction of GAF. “whose function remains enigmatic“ is just a rhetoric statement, as if now the function would be clear, how should a backstop to pivoting look like?

Based on the flex we see between Pr and Pfr for PsBphP1, we are still led to speculate that the PAS domain could provide rigidity to the PSM thus focusing its movements toward the OPM. We agree that the OPM region also provides stability to the dimer.

As for the pivot, the pivot point that we are seeing is not a speculative pivot. It is an actual pivot. We have also seen this with various crystal structures of DrBphP, which can be found from available structures in the PDB, which suggests it is more universal than just PsBphP1. However, the dimer interfaces of DrBphP and PsBphP1 at the GAF domain are strong, which could suggest Agp1 is governed by a slightly different mechanism. The situation with Agp1 is currently unknown, since there are no paired Pr/Pfr structures of the full-length molecule. It is possible that Agp1 behaves differently. Obligate dimerization of the HK domains would impose a very high local concentration of GAF domains, which would boost the probability that they dimerize. Through that process they may undergo very similar photostate-dependent pivots at the GAF domain interface. We would suggest that the reviewer tests this with Agp1.

As for the PAS domain, it may simply provide a barrier to ensure a more precise dimer interface. This is speculative, but since the interface is hydrophobic, it would help limit the number of binding modes by steric hindrance, much like a door stop. Here, the wording was changed to say: **“Our speculation is that this pivot buffers large motions at the PHY domain from dissociating the dimeric interface at the GAF domains, so that conformational energy can instead be directed toward the HK bidomains.”**

- 326-330 what is the evidence for this assumption? Note that many bacterial phytochrome histidine kinase are down regulated upon Pr to Pfr transition. This should be mentioned in the introduction and considered in the discussion when it comes to universal statements.

We have reworded the sentence on lines 359-360 to read: **“BphPs with transmitter kinase activity (at least those with accelerated activity as Pfr)....”**. We agree that there are a number of bacterial Phy kinases that are more active as Pr (not Pfr). How photoconversion affects them is unclear.

- 332-337 DoD .. telling about the mechanism, sorry, I did not understand what is meant.

For a kinase to work, it needs to bind the substrate in a transient fashion to allow more than one reaction cycle. The kinase cannot be tightly bound to the response regulator, otherwise the reaction would stall after one event. For PsBphP1, the CA domains bind relatively tightly in an unproductive position as Pr. In Pfr, the CA domains are released and now able to interact with the phosphoacceptor histidine.

- 337 why is it a key aspect that contact is transient?

By SEC the contact between *PsBphP1* and *PsAlgB* is imperceptible. Since the phosphorylation reaction is rapid, it is likely that the interaction between the CA domain and the DHp domain near the histidine is weak. Moreover, the facts that DHp domains help form DoDs and two of the CA domains have no associated EM density suggest that the interaction between CA domains and DHp domains is very weak. Here, if there was moderate affinity of the CA domain for the DHp, by the law of mass action the expectation would be high CA domain occupancy at the DHp domain. Instead, we see DHp domain association to the exclusion of CA domain binding. In other words, it is apparent that the interaction between the CA and DHp domains is very fleeting and transient.

- 351 avoid such challenges, was this really the motivation? Without fusion would rather be the standard approach.

Reworded Lines 384-385 to say: **“To avoid the inherent challenges that translational fusions present, we attempted to assemble a native *PsBphP1*/response regulator complex without covalent coupling.”**

- 362 and following, if I understand it right, there is no specificity for response regulator. On the other hand, it is often spoken of cognate response regulators, kinase and regulator are in the same operon. What about *Pr-Pfr* in trans phosphorylation (see below)?

There is clear specificity to histidine kinase/response regulator pairs, otherwise none of them would work as planned. As per the organization of many operons in bacteria, many related proteins are clustered and co-expressed, thus for example providing indirect evidence that *PsBphP1* and *PsBphR* work as pairs. We do not know about transphosphorylation for heterodimers in the case of *PsBphP1*. *PsBphP1* interacts with *PsAlgB* as evidence by phosphorylation, but it does not remain bound.

- 365 was the His-tag at the N-terminus? Fig. S1, where is the Gly remnant?

Yes, we added on line 548 **“N-terminal”** for clarification.

- 367 EM was made for *Pr Pfr*, were there any *Pr Pfr* comparisons with respect to autophosphorylation or phosphotransfer?

Prior published studies by us already showed that *PsBphP1* has poor kinase activity as *Pr* (Bhoo et al., 2001 Nature; Li et al., 2022 Nature; Burgie et al. 2023 Nat. Plants). We did not attempt to make and test *Pr:Pfr* heterodimers.

- 373, to accept *P* is certainly not dependent on the strength of labeling.

Reworded sentence on lines 407-409 to read: **“In the latter case, the *Asp59-Ala* mutant of *PsAlgB* failed to accept ³²P even in the presence of highly labeled, wild-type *PsBphP1*.”**

- 379 SEC can be used to detect protein interactions, if they are strong. How strong does an interaction have to be for SEC to be detected? In other words, if there is no effect in SEC, it does not mean that there is no interaction. For *Agp1/Agp2* interaction we found: no effect on SEC, effect on photophosphorylation, effect on spectra, effect on FRET. The statement „failed to bind“ is not correct, because it is not known that it does not bind. Maybe „failed to affect mobility of *BphP* on SEC columns“. Please check also abstract. See also 384.

There is clearly an interaction between *PsBphP1* and *PsAlgB* to enable phosphotransfer just not strong enough to allow co-elution by SEC. We do not know how stable an interaction needs to be for SEC co-elution but it is likely concentration dependent.

- 394 interesting finding on the stimulation of phosphorylation , it would be good if it could be investigated further.

Agreed, likely in the next study.

- 416 ZZZssa to ZZEssa. Here comes a mixture of known results and the present data. This is ok for the discussion but should also be named as such (line 413 ..) . I would prefer if would be clear what was already known or suggested and what is new. For example that first step is the isomeriation it clear, but is not the result of the present work.

We think that the paragraph on lines 454-464 is clear as written. It was crafted to state what we observed by comparing Pr and Pfr models of PsBphP1 how many photoconversion events were similar to those seen previously. The sentence before provided the historical context along with references.

- 418 selection between two rotational orientations of the PHY domain. I do not have in mind whether this is described in the results section, but it could be explained here again. - 419 prying force .. translates .. helical spine : where is the prying force, what kind of motions. Which feature spans via ... , the prying force, the translation', the movement of helices? Could be more precise.

We clarified this section on Lines 459-462 to read: **"...translations driven by the hairpin as it converts from antiparallel β -stranded as Pr to α -helical as Pfr, which forces the downward rotational pivot of the PHY domain. The resulting prying force enabled by the structural stability of the PHY domain translates..."**

- 430 why is it an advantage of EM, there are not necessarily different forms.

The advantages of cryo-EM are many, including not needing to generate crystals and adaptable to larger proteins and protein complexes. And as illustrated here, you can see multiple populations by cryo-EM which is usually not possible for proteins locked in a crystal lattice.

- 441: adjustment, are they not adjusted? Is there evidence that dimer contacts need to be adjusted? Why complete dissociation? For me, the sentence was distracting.

Changed on line 486 to say: **"...included variations in the observed..."**

- 460, one would like to hear some words here or elsewhere that many bacterial phytochrome histidine kinases have a strong autophosphorylation in the Pr and weaker in the Pfr.

We added a statement to about its kinase activity and references on line 471-473. Since we have little understanding of how Pr-specific Phy kinases might work since they would be theoretically active in the dark-adapted state, we have avoided any speculation of this subgroup.

- 461 helps explains → helps explain.

Fixed

- 466 a second what, first is missing.

We added **"The second peculiarity is...."** to Line 521-522 in new paragraph for clarification.

- 505 why does decrease in temperature result in enhancement of heme synthesis, even before HO is induced?

From our years of experience, we find that lowering the temperature gives us better yields of photoactive phytochromes. As an FYI, the HO enzymes break down heme and do not direct its synthesis.

- 507 also here, lowering of temperature for induction of expression?

Same benefit as above.

- 510 funny, but does your centrifuge have green light inside? Centrifugation force and time or refer to a previous work.

We made the process more clear on Line 565.

- 514 fourfold of what.

We made the extraction process more clear of Lines 568-569.

- 516 518 was PMSF added twice?

We removed second PMSF, thanks.

- 524 diameter and length of Sephadex is important, also the loading volume. For good separation, loading volume should be 1% of column volume, this would be 2 ml. For buffer exchange, more is possible but in any case, one would like to know the volume.

This step is obviously for buffer exchange, thus the column should be around 4 times the sample volume. For accuracy, we added “~25 mL” to the description of the column volume on line 578.

- 530 good luck with the Q Sepharose, in our case it denatured the phytochromes.

This column worked for us.

- 532 and 533, what was the protein concentration.

- up to 556, would be good if protein concentrations would be given more often.

- 560 and 571, different concentrations given for the same sample.

For the three questions above, we included the culture volumes, sample volumes, or protein concentrations when necessary for others to repeat the protocol. We did not measure protein concentrations at every purification step for every single preparation, but used SDS-PAGE to follow sample purity. You should be able to follow the procedures with the indicated numbers now included in the Methods.

- 479 detectable.

We corrected the spelling error.

- 483 remove „a“.

We removed the “a”.

- 622 DoD PSM.

We inadvertently duplicated this and the next paragraph, which was fixed by removing the first paragraph. The second paragraph starts on line 675-683.

- 624 check for double blanks.

As a standard practice, we use two spaces between sentences.

- 680 to get a general impression of HK folding: were also other HK structures compared? How many are there in the database? How similar is the second hit? Etc. In general, the folding should be better explained in the results section: what parts were folded.

The paragraph on lines 723-727 was modified as follows, **“This model matched closely to CA or CA-like domains from over 100 proteins from the RCSB Protein Data Bank as determined by the DALI server (ekhidna2.biocenter.helsinki.fi/dali)³². To illustrate this congruity we superposed the predicted PsBphP1 CA domain with the X-ray crystallographic 3D model of the ADP-bound CA domain from *T.***

***maritima* HK853 available in the under PDB ID code 4JAV³¹. “**

For clarity we did not superpose the model with multiple CA domains, because we did not want to make the 2D-image, too confusing. The residues used for the prediction were already listed in the second sentence.

- 690, loading volume.

- 706 column and sample volumes? As above or different? Which was used when?

As stated above, we added details to the purification protocol for those that seemed critical for replication.

- 711 fixing solution should contain acetic acid, in methanol only, the proteins can diffuse, but ok, apparently it worked.

Methanol is also a fixing agent by itself and it did work. Acetic acid was omitted for fear that the low pH might release the phosphates from the histidines or aspartates.

- 718 for what was Excel used and for what R.

Clarified on line 760 to read: “**Data analyses were conducted using the R software package (<https://www.r-project.org>).**”

- 731 volume of sample and column dimensions important.

The Superdex 200 Increase 10/300 GL column is a column of defined volume. 10 mm x 300 mm. “**100 µL samples**” was added to lines 774-775.

- 736 what is Unicorn.

Its from **Cytiva**. Now added to the Methods section on line 779.

- Comments on suggested references from the Lamarter lab.

We did survey the list of possible references provided by Review #2 as appropriate for this publication. Notably, reference [1] by Nagano et al (2016) in JBC was most useful and we added it at several places in the paper. Thanks for the suggestion. As for the others, they did not seem as appropriate as compared to the references we used to support specific points and were not added. Several of these suggested references used protein analysis techniques that were not as clarifying as actual 3D structures that now exist for several Phys, or investigated a Phys with unusual enzymatic activity, that being *Synechocystis* Cph1 that behaves as a kinase opposite to that of *PsBphP1* by having robust activity in the Pr and not Pfr form.

Reviewer #3 Comments:

*Phytochromes are a superfamily of red/far-red photosensing proteins in diverse species and regulate downstream signaling pathways. The manuscript by Burgie et al. aims to elucidate the mechanistic basis of photoconversion by solving the resting (Pr) and the photoactivated (Pfr) structures of a bacteriophytochrome. To this aim, the full-length, biliverdin-bound bacteriophytochrome BphP1 from *P. syringae* (PsBphP1) was expressed and purified. Structures at overall resolutions between 2.8 Å and 3.4 Å of the Pr and Pfr states were obtained. The authors provide based on these structures, a mechanism how the configuration of the biliverdin allosterically translates to the interconnected histidine kinase (HK) domain, thereby increasing its autokinase activity and affecting the cognate response regulator AlgB. For the Pfr state, medial, splayed and tetrameric (dimer-of-dimer) configurations were obtained. For the Pr, a single conformation was obtained. Unresolved/not interpretable features within obtained Coulomb potential maps, e.g., the catalytic ATPase domains (CA), were modelled by a predicted AlphaFold model, providing a hypothetical structure of the dimeric full-length protein.*

Numerous structures of BphPs (although some are only fragments) were published previously and an allosteric communication mechanism was reported (PMID: 36509762) raising serious questions about the novelty of the here presented work. Furthermore and (also) importantly, I have serious concerns related to the resolution of the maps and the subsequent interpretation. While the photosensory modules (PSMs) are well resolved, as in previously published studies, the placement and structures of the effector domains rely on predicted, hypothetical models and fitting into non-existing (Fig. 4) and very low resolution density maps. This further weakens the claims and the novelty considerably.

Major points from above in more detail

- Since this manuscript does not provide the very first complete structures of a phytochrome in both Pr and Pfr states (the first reported is the phytochrome from *Deinococcus radiodurans* in 2022, PMID: 36509762), its novelty is limited. The structural consequences of the photoconversion in the PSM part is similar as reported previously.

We agree that a prior structure of a Phy-response regulator (RR) fusion by Wahlgren et al. (2022 *Nat. Commun.* (PMID: 36509762)) was the first to attempt to report a full-length dimeric structure of a Phy that acts within a transmitter kinase cascade by fusing it to its predicted response regulator *DrBphR* in an attempt to stabilize the HK domain. However, it should be clarified that *DrBphR* acts as a phosphatase (at least *in vitro*) in association with *DrBphP* with stronger activity seen for Pfr not as a phosphoacceptor (at least *in vitro*). In the *DrBphP* model Wahlgren et al. studied, they simply fused the phosphatase to the C-terminus of the HK domain via a 12-amino-acid tether. While they could show that the fusion functioned as a phosphatase through dephosphorylation assays of *DrBphR*, they could not tell whether this removal was in *cis* or *trans* in the reaction mixture (i.e., removing the phosphate from its own dimer or even own protomer, or removing the phosphate from other dimers). Nor could they determine whether the *DrBphR* was correctly positioned within the complex nor whether it compromised folding of the HK domain from *DrBphP*. One can easily image that this ectopic covalently tether could prevent normal movement of the CA domain either by the tether or by virtue of being bound to its end. As can be seen in the resulting models by Wahlgren et al., the positions of *DrBphR* seem to be weirdly positioned at the C-terminus of *DrBphP* likely reflecting this non-native covalent linkage.

In support, our comparisons of the published Wahlgren et al. Pr and Pfr structures by superposition revealed little movement in the *DrBphP* photoreceptor upon photoconversion except for the hairpin transition. By comparison, we saw more widespread changes upon superposition of the Pr and Pfr structures for *PsBphP1* (see Figure to the left). We added a statement to this effect on lines 505-507.

One obvious possibility is that the changes that normally occur during in *DrBphP* photoconversion were suppressed by using a fusion protein.

This possibility of a strong artifact is why we did not use this fusion approach and stated so in the beginnings of our search for the cognate response regulator for *PsBphP* which ultimately led to the discovery of AlgB as *PsBphP1*'s reaction partner (see lines 375-385). The Wahlgren et al. structure of *DrBphP* is also complicated by linking the phosphatase activity of *DrBphR* to *DrBphP* versus linking the phosphorylation target of *DrBphP* (which is currently unknown) and is actually one of the reasons why we switched to using *PrBphP* as a model and searched for its phosphorylation target. See Lines 384-395.

As for assembly of the full model of *PsBphP1*, we hope that the discussions above along with detailed explanations of map assembly provide sufficient confidence in the final models. If wanted, we could include the paired superposed models of *DrBphP* and *PsBphP1* shown above in a Supplementary Figure to illustrate potential complications with studying ectopic fusions.

- While the EM density of PSM and minor part of the S-helices are reasonable for the built model and made interpretation, care must be taken with overinterpretation of the weak/absent density and following modelling with AlphaFold models. This yields hypothetical 3D models of the different states. Therefore the made claims are ambiguous and problematic, e.g., Fig. 4.

We agree that over-interpretations of low resolution models are a real problem. Which is why we also combined both RoseTTAFold and AlphaFold predictions with these lower resolution EM maps.

- The finding that PsBphP1 as Pfr forms tetramers is remarkable and its dynamic equilibrium was subject of investigation. Unfortunately, it's very unclear/questionable from a technical standpoint how this map was obtained. Regarding the workflow in Suppl. Fig. 4: It seems as if the tetramer map was generated from symmetry-expanded dimer map with no ab-initio model generated before, which is problematic. As a minor issue, the local resolution for the tetrameric map as well as the orientation distribution plots for all final maps are not indicated.

As also requested by Reviewers #1 and #2, we described how the models were generated more explicitly in the Abstract, Introduction, Results, and Discussion. As examples see lines 36-40 in the Abstract, lines 110-121 in the Introduction, and lines 216-220; lines 229-234, and lines 300-308 in the Results. We also added the cryo-EM map of the DoD interface along with aligned models of the DHP and CA domains to demonstrate the accuracy of the modelling in Supplementary Figure 10

- While previous structures are referenced, there is no comparison (except the BV overlay in Fig. 1b) between them and the structures solved here.

Given the already large number of Figures and Supplementary Figures, we avoided such comparisons except in Fig. 1b for the bilin and a comparison of PHY domains from 51 available structures in Supplementary Fig. 10. We did state on Lines 170-171 the similarity of our structures to previous reports.

“The resulting high-resolution Pr model conformed well with prior PSM structures of BphPs^{11,12,14,15,22,29-31} (Fig. 1d,e).”

- Supplemental Fig. 3, panel c is misleading. The attributed resolution of 3.3 Å is difficult to understand considering the red density parts that would be expected to be less resolved than 4.5 Å. It would be important to show the mask used to obtain the indicated 3.3 Å.

We thank Reviewer #3 for addressing this ambiguity. For clarity, we updated panel c of Supplementary Fig. 3 to include the mask used for the 3DVA, local refinement, and local resolution estimation. The map shown is unsharpened and contoured at a lower threshold value so that the density of the DHp domains could be seen. We also revised the figure legend for Figure 3 as follows:

“(c) 3D variability analysis (3DVA) followed by local refinement generated a 3.3-Å map for 119,594 particles that included additional residues within the DHp domains (PDB 8U8Z). Shown is the unsharpened final map with the dilated mask used for the 3DVA, local refinement, and local resolution estimation. The mask had an additional soft padding width of 20 pixels.”

Less major and minor points:

- *What is the rationale behind using the chaotropic reagent thiocyanate as a stabilizer in the “enhanced solubility buffer”? I agree that on a global scale the structure seems not to be affected, but locally it could disrupt interactions and contribute to higher mobilities of certain domains.*

As mentioned in the paper on Lines 199-203, our initial cryo-EM of full-length *PsBphP1* revealed aggregation that likely complicated blob picking of relevant particles of the dimer in the EM grids. To help avoid this issue, we tried relatively low concentrations of thiocyanate (175 mM) to improve solubility. While we cannot definitely say that local features of the photoreceptor were unaltered, we compared preparations in KCl versus KSCN based on: (1) Pr and Pfr absorption spectra and maxima, (2) spectral change ratios of Pr/Pfr difference spectra, (3) Pr to Pfr photoconversion rates, (4) stability of the photoreceptor based on thermal reversion of Pfr back to Pr, (5) autophosphorylation rates as Pfr, and (6) negative staining EM. (see Supplementary Figures 1 and 2). None of these assays revealed strong effects on the photoreceptor. That the PSM structure seen here by cryo-EM in KSCN resembled that seen for other Phys by X-ray crystallography also supports this point. It should also be mentioned that low levels of chaotropic agents and detergents are common additives for cryo-EM before sample freezing to help with protein solubility and the randomization of particle orientations

- *Suppl. Fig. 2: To pretend that KSCN has no effect on the 3D structure of the *PsBphP1* protein based on negative stain data (low resolution, heavy metal salt and specimen drying) and 2D class averages is not credible.*

See the points articulated above on this point of potential problems in using KSCN. As described on lines 139-153, we went to great lengths to address potential problems with using KSCN. Maybe they were missed by Reviewer #3.

- *The Reviewer suggest careful reading of the manuscript, here a few issues:*

- *line 518: PMSF is mentioned twice.*

We removed the second PMSF.

-- *line 239: A bracket is missing.*

We added the bracket.

-- *line 281: punctuation mark.*

We removed the period.

-- *Inconsistency: e.g., cryo-EM in line 231, cryoEM in line 430.*
We fixed the paper to say “cryo-EM” through-out.

-- *line 150: 2D class averages cannot be collected – please rephrase.*
The sentence on Line 159-160 now reads “**we collected an ensemble of highly resolved 2D class averages**”.

-- *Fig. S1: remove the text “Lorem ipsum” ...*
We fixed this weird error.

-- *“Supplemental” should be replaced by “Supplementary”*
We fixed this spelling throughout the manuscript.

-- *The authors might want to revisit model building: For example, in Supplemental Fig. 6, panel b, residues of S-helix(B) don't seem to fit well within the density.*
From our vantage point, the S-helices in Supplementary Fig. 6b do seem to fit well for as far as they could be modeled in the Pr model (PDB 8U4X) and the DoD Pfr model (PDB 8U62).

In conclusion, we thank the editor and Reviewers #1-3 for help in improving this manuscript. We hope it is now ready for publication in *Nature Communications*.

Sincerely,

Dr. Richard D. Vierstra
George and Charmaine Mallinckrodt Professor
Department of Biology, Campus Box 1137
Washington University in St. Louis
One Brookings Drive,
St. Louis, Missouri 63130 USA
(314)-935-5058 office;
(314)-935-4432 fax
rdvierstra@wustl.edu
<https://biology.wustl.edu/people/richard-d-vierstra>

REVIEWER COMMENTS

Reviewer #1 (Remarks to the Author):

The authors addressed all of the concerns I had in the initial version of the manuscript, substantially improving it.

One minor issue: (line 820) "Cryo-EM data were collected on a ThermoFischer Titan Krios microscope" -> edit to "...Thermo Fisher.."

Reviewer #2 (Remarks to the Author):

My comments to the first version were all considered and in most cases handled in proper way. In line 457, ZZE should be ZZZ.

Reviewer #3 (Remarks to the Author):

I appreciate the author's efforts to improve the manuscript, in particular the clarifications throughout the manuscript that the models presented were generated from a combination of experimental (cryo-EM) and structural prediction (RoseTTAFold and AlphaFold) results. This combination was also questioned by the other reviewers.

At this point, I wish to express my reservations about the 'patchwork' structure provided, particularly because the novel and most interesting domains were determined at an unsatisfactory, low resolution that precludes accurate model building.

For the sake of transparency, we kindly request the authors to address and modify the following points:

- The maps colored by the local resolution estimates should cover the whole - or at least a reasonable - resolution range. This will support the reader to clearly identify problematic regions (currently the local resolution is limited to 4.5 Å (Figs. S3 and S4) or 7 Å (Fig. S5)).
- I would like to point out that the enforcement of C2 symmetry in the (considerably asymmetric, most likely due to the flexibility of PSM domains relative to the HK domain) low-resolution DoD map in Fig. S5 is problematic. Please add a statement in legend of Fig. 3b that this is a symmetrized volume.
- The authors might want to look carefully at the FSC plot (which crosses the FSC = 0.143 criterion twice) at the very bottom of Fig. S5. A more soft mask might resolve this issue.
- Line 258: For consistency, add a hyphen in "3.3 Å" (or delete them everywhere)

Reviewer #4 (Remarks to the Author):

May 3, 2024

Response to Reviewers- NCOMMS-24-00462B

"Signaling by a Bacterial Phytochrome Histidine Kinase Involves a Conformational Cascade Reorganizing the Dimeric Photoreceptor"

E, Sethe Burgie, Kathrine Basore, Michael J. Rau, Brock Summers, Alayna J. Mickles, Vadim Grigura, James A. J. Fitzpatrick, and Richard D. Vierstra

Overall Response:

We were pleased to read such favorable reviews of the revised manuscript. Hopefully, we covered most, if not all, of the remaining outstanding questions made by Reviewer #3 in this updated version. We particularly want to thank Reviewer #3 for the advice of using a soft mask for final resolution of the HK region in the DoDs as Pfr. This fixed the strange FSC blot in Supplementary Figure 5 and improved the map resolution from 6.7 to 4.1 Å. While the EM map did not change, our confidence in the map certainly improved.

REVIEWER COMMENTS

Reviewer #1 (Remarks to the Author):

The authors addressed all of the concerns I had in the initial version of the manuscript, substantially improving it.

One minor issue: (line 820) "Cryo-EM data were collected on a ThermoFischer Titan Krios microscope" -> edit to "...Thermo Fisher..."

Fixed as recommended.

Reviewer #2 (Remarks to the Author):

My comments to the first version were all considered and in most cases handled in proper way. In line 457, ZZE should be ZZZ.

Fixed as recommended.

Reviewer #3 (Remarks to the Author):

I appreciate the author's efforts to improve the manuscript, in particular the clarifications throughout the manuscript that the models presented were generated from a combination of experimental (cryo-EM) and structural prediction (RoseTTAFold and AlphaFold) results. This combination was also questioned by the other reviewers.

At this point, I wish to express my reservations about the 'patchwork' structure provided, particularly because the novel and most interesting domains were determined at an unsatisfactory, low resolution that precludes accurate model building.

For the sake of transparency, we kindly request the authors to address and modify the following points:

- The maps colored by the local resolution estimates should cover the whole - or at least a reasonable - resolution range. This will support the reader to clearly identify problematic regions (currently the local resolution is limited to 4.5 Å (Figs. S3 and S4) or 7 Å (Fig. S5)).

We thank the reviewer for this suggestion. The ranges of local resolution used for map coloring have been changed where appropriate.

- *I would like to point out that the enforcement of C2 symmetry in the (considerably asymmetric, most likely due to the flexibility of PSM domains relative to the HK domain) low-resolution DoD map in Fig. S5 is problematic. Please add a statement in legend of Fig. 3b that this is a symmetrized volume.*

We made the following recommended addition to the legend of Fig 3 on lines 1038-1039:

C2 symmetry was applied to the map prior to symmetry expansion and local refinement (see Supplementary Fig. 5).

- *The authors might want to look carefully at the FSC plot (which crosses the FSC = 0.143 criterion twice) at the very bottom of Fig. S5. A more soft mask might resolve this issue.*

We thank the reviewer for this suggestion. We generated a slightly softer mask, which fixed the issue. The mask used and the updated FCS plot were added to Fig. S5, and the following edits were made to the legend on Lines 1220-1222 (changes are highlighted).

“To further refine the DHp domains, a focused mask was applied during a subsequent round of local refinement. The dilated mask shown had an additional soft padding width of 10 pixels. Resolution anisotropy was noticed, as evidenced by the increased correlation bump at higher resolution, and was likely generated by the preferred orientations of the particles. The final map at 4.1-Å resolution was colored-coded based on the local resolution.”

- *Line 258: For consistency, add a hyphen in “3.3 Å” (or delete them everywhere)*

As a standard practice, we hyphenated these terms when they are adjectives supporting a noun (e.g., 3.3-Å structure). In this case, there is no noun.

Reviewer #4 (Remarks to the Author):

Thanks for your participation.

In conclusion, we thank the editor and Reviewers #1-3 for help in improving this manuscript. We hope it is now ready for publication in *Nature Communications*.

Sincerely,

Dr. Richard D. Vierstra
George and Charmaine Mallinckrodt Professor
Department of Biology, Campus Box 1137
Washington University in St. Louis
One Brookings Drive,
St. Louis, Missouri 63130 USA

REVIEWERS' COMMENTS

Reviewer #3 (Remarks to the Author):

The authors addressed the concerns I had with the previous version of the manuscript, considering and largely implementing them.